# Acidic C-terminal domains autoregulate the RNA chaperone Hfq

Andrew Santiago-Frangos[1], Jeliazko R Jeliazkov[2], Jeffrey J Gray[3], Sarah A Woodson[4]*

[1]Cell, Molecular and Developmental Biology and Biophysics Program, Johns Hopkins University, Baltimore, United States; [2]Program in Molecular Biophysics, Johns Hopkins University, Baltimore, United States; [3]Department of Chemical and Biomolecular Engineering, Johns Hopkins University, Baltimore, United States; [4]T.C. Jenkins Department of Biophysics, Johns Hopkins University, Baltimore, United States

**Abstract** The RNA chaperone Hfq is an Sm protein that facilitates base pairing between bacterial small RNAs (sRNAs) and mRNAs involved in stress response and pathogenesis. Hfq possesses an intrinsically disordered C-terminal domain (CTD) that may tune the function of the Sm domain in different organisms. In *Escherichia coli,* the Hfq CTD increases kinetic competition between sRNAs and recycles Hfq from the sRNA-mRNA duplex. Here, *de novo* Rosetta modeling and competitive binding experiments show that the acidic tip of the *E. coli* Hfq CTD transiently binds the basic Sm core residues necessary for RNA annealing. The CTD tip competes against non-specific RNA binding, facilitates dsRNA release, and prevents indiscriminate DNA aggregation, suggesting that this acidic peptide mimics nucleic acid to auto-regulate RNA binding to the Sm ring. The mechanism of CTD auto-inhibition predicts the chaperone function of Hfq in bacterial genera and illuminates how Sm proteins may evolve new functions.
DOI: https://doi.org/10.7554/eLife.27049.001

*For correspondence:
swoodson@jhu.edu

Competing interests: The authors declare that no competing interests exist.

## Introduction

Host factor for RNA phage Qβ replication (Hfq) is found in most sequenced bacterial genomes (*Sun et al., 2002*) and plays a well characterized role in post-transcriptional regulation by small non-coding RNA (sRNA) (*Gottesman et al., 2006*; *Storz et al., 2011*). Regulation by Hfq and sRNAs is important for controlling the expression of metabolic, stress-response and virulence genes in many genera (*Feliciano et al., 2016*). Hfq binds sRNA and facilitates interactions between sRNAs and their mRNA targets (*Zhang et al., 2002*; *Moller et al., 2002*). To chaperone sRNA target recognition, Hfq must select its substrates from a large pool of nucleic acid in the cell and efficiently dissociate from its products at the end of each RNA annealing cycle (*Rajkowitsch et al., 2007*).

*E. coli* Hfq contains an Sm-like domain (residues 7–65) that oligomerizes into a homohexameric ring with two sequence-specific RNA-binding faces. The proximal face of the ring is highly conserved and binds to uridines (*Zhang et al., 2002*; *Schumacher et al., 2002*) at the 3'-ends of bacterial small non-coding RNA (sRNA) that resemble a classic Sm binding site (*Zhou et al., 2014*). In *E. coli* and many Gram negative bacteria, the distal face of Hfq binds to AAN triplet repeats (*Mikulecky et al., 2004*; *Link et al., 2009*) found in mRNA leaders (*Link et al., 2009*; *Soper et al., 2011*) and certain sRNAs (*Schu et al., 2015*; *Małecka et al., 2015*). In addition to these sequence-specific RNA binding sites, arginine-rich basic patches at the rim of the *E. coli* Hfq hexamer interact with the sRNA body (*Zhang et al., 2002*; *Otaka et al., 2011*; *Sauer et al., 2012*; *Ishikawa et al., 2012*; *Zhang et al., 2013*) and facilitate annealing with target mRNAs (*Panja et al., 2013*; *Zheng et al., 2016*).

Like many RNA binding proteins, Hfq also possesses intrinsically disordered domains that have the potential to modulate the function of the core Sm ring. The *E. coli* Hfq Sm domain is flanked by a short, disordered, N-terminal domain (NTD; residues 1–6), which protrudes from the proximal face of the hexamer, and a longer disordered C-terminal domain (CTD; residues 66–102), which extends from the rim (*Beich-Frandsen et al., 2011a*; *Vincent et al., 2012*). NMR chemical shift perturbations from a comparison of full-length Hfq (Hfq102) and a truncated variant lacking the CTD (Hfq65) suggested that some part of the CTD contacts residues on the rim of the hexamer, although the specificity of these proposed contacts was uncertain since they occur near where the CTD protrudes from the ring (*Beich-Frandsen et al., 2011a*; *Vincent et al., 2012*).

The functional importance of the CTD for sRNA regulation has also been unclear, owing to the conflicting results of different studies (*Sonnleitner et al., 2004*; *Olsen et al., 2010*; *Večerek et al., 2008*; *Salim et al., 2012*). Using a combination of biophysical and genetic approaches, we recently showed that the CTD displaces RNA from the rim and proximal face of Hfq (*Santiago-Frangos et al., 2016*), with two important consequences. First, release of annealed dsRNA from the arginine-rich rim is accelerated, increasing Hfq turnover. Second, kinetic competition between sRNAs is increased, which allows dominant sRNAs to bind to Hfq and accumulate in the cell, while weaker competitors are degraded (*Santiago-Frangos et al., 2016*). The latter creates a hierarchy of sRNA regulation that depends on the CTD.

The mechanism by which the CTD displaces RNA from the core (Sm domain and NTD) of Hfq is unknown. No common sequence motifs have been identified in the CTD (*Sun et al., 2002*; *Vincent et al., 2012*; *Weichenrieder, 2014*; *Sobrero and Valverde, 2012*; *Fortas et al., 2015*; *Updegrove et al., 2016*), which varies in length and composition across bacteria (*Attia et al., 2008*; *Schilling and Gerischer, 2009*; *Baba et al., 2010*). This diversity is characteristic of disordered peptides, which rapidly evolve via non-conservative substitutions and indels (*Liu et al., 2008*; *Brown et al., 2010*; *Light et al., 2013*). Two models could explain the displacement of RNA by CTDs in *E. coli* Hfq. The 'polymer brush' model suggests the CTDs passively obstruct RNA binding sites. This model is attractive since it depends only on the length and flexibility of the CTD. In contrast, the 'nucleic acid mimic' model suggests that the CTDs specifically bind to basic core residues and actively compete against nucleic acids. Given the divergence of CTD and core sequences, this model predicts that CTD auto-regulation is likely in some Hfq clades but not others.

In this study, we use *de novo* modeling and biophysical experiments to determine the mechanism by which the CTD regulates Hfq activity. We propose that the acidic CTD tip transiently binds the rim of *E. coli* Hfq and makes distributed interactions with basic residues, thereby modulating RNA and DNA binding and RNA annealing. Applying our modeling procedure to Hfqs from other bacteria demonstrates that stable interactions between the acidic CTD and the basic rim correlate with the importance of Hfq for sRNA regulation in that host. Thus, our proposed mechanism of CTD auto-inhibition provides a basis for predicting the function of Hfq in different bacteria. Our approach may be useful for predicting the sequence–function relationship of disordered domains in other partially disordered proteins.

## Results

### C-terminus of Hfq is enriched for acidic residues

To search for conserved features or amino acid motifs amongst the highly heterogeneous Hfq CTDs, we first built a phylogenetic tree (*Figure 1—figure supplement 1*) from the multiple sequence alignment of nearly 1000 non-redundant sequences (see Methods). The cluster containing *E. coli* Hfq contained many other Hfq variants previously identified as functional in RNA annealing (*Zheng et al., 2016*) or sRNA regulation (*Gottesman et al., 2006*). Therefore, we examined the sequence logo of this cluster of 222 Hfqs in more detail (*Figure 1A*).

The start of the CTD region is delineated by a proline at position 64 of *E. coli* Hfq that is strongly conserved across all clades. Additionally, an arginine at the beginning of the CTD (position 66 in *E. coli*) that packs against the lateral edge of the Hfq hexamer (*Beich-Frandsen et al., 2011a*; *Sauter et al., 2003*; *Dimastrogiovanni et al., 2014*) is strongly conserved. Although the middle linker region of the CTD lacks conserved motifs (*Figure 1—figure supplement 1*), the C-terminus is rich in acidic residues, corresponding to the sequence DSEETE in *E. coli*. Noting that most Hfq

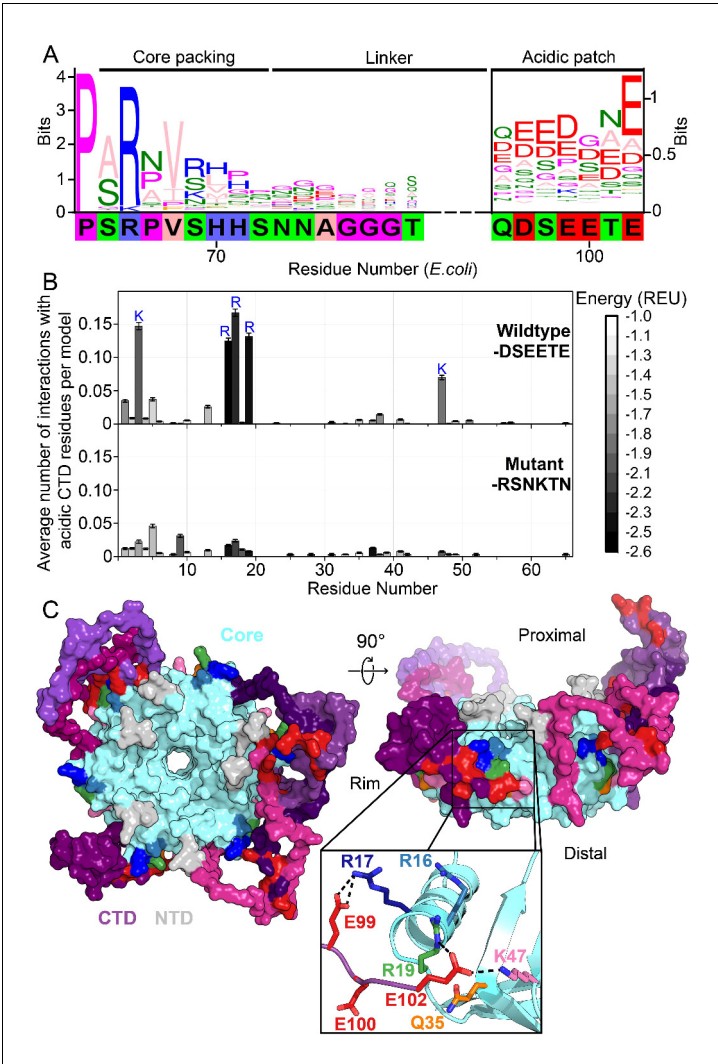

**Figure 1.** Acidic residues in the Hfq CTD are predicted to bind basic core. (**A**) Sequence logo (*Crooks et al., 2004*) of the CTD generated from gapped alignment of Hfq sequences that clustered with *Escherichia coli* Hfq (Group 1 in *Figure 1—figure supplement 1A,B*), numbered according to the *E. coli* sequence. Regions of interest are denoted above. The gapped *E. coli* CTD sequence is shown below. Eukaryotic Sm proteins cluster separately (*Figure 1—figure supplement 1D*). (**B**) (Top) Average number of times a given core residue favorably interacts ($E < -1.0$ Rosetta Energy Units) with at least one acidic CTD residue, per low energy model. Acidic CTD residues most frequently interact with basic Hfq core residues. (Bottom) Mutation of acidic CTD residues 97, 99, 100 and 102 to basic or polar residues decreases the number of predicted core interactions. Error bars represent ± 1 s.d. as computed by bootstrap resampling of the computational models (see Methods and *Figure 1—figure supplement 2*). Of 36 core residues not predicted to interact with the CTD, 14 had accessible surface area < 2.0 $Å^2$, computed in PyMOL. (**C**) (Left) Example low-energy model of wildtype *E. coli* Hfq; top-down proximal view. Light grey, NTD; cyan, Hfq core; pink-purple, CTD; red, CTD tip. (Center) Side view of rim of the same Hfq model. (Inset) Example hydrogen bonding network at the CTD–core binding interface showing interactions between the acidic CTD residues (red) and core residues as indicated. Additional models in *Figure 1—figure supplement 3*.

DOI: https://doi.org/10.7554/eLife.27049.002

The following figure supplements are available for figure 1:

**Figure supplement 1.** Alignment and clustering of Hfq sequences used in this study.

DOI: https://doi.org/10.7554/eLife.27049.003

**Figure supplement 2.** Semi-log plot of the lowest observed energy for a given number of attempted low-resolution backbone moves.

DOI: https://doi.org/10.7554/eLife.27049.004

**Figure supplement 3.** Modelled CTD-core interactions are heterogeneous.

*Figure 1 continued on next page*

*Figure 1 continued*

DOI: https://doi.org/10.7554/eLife.27049.005

clusters containing a basic patch on the rim also end in acidic residues, we hypothesized that the CTD tip binds the rim. Because the basic patch is essential for sRNA binding and annealing, direct interaction between the CTD tip and the Hfq core could explain the previously observed auto-inhibition of the CTD (*Santiago-Frangos et al., 2016*).

## De novo modeling of CTD interactions in the Hfq hexamer

To determine whether the acidic tip of the *E. coli* Hfq CTD could interact with basic residues in the core, we used Rosetta FloppyTail (*Kleiger et al., 2009*), a *de novo* modeling approach for disordered regions of proteins. We updated the original FloppyTail algorithm to model multiple disordered regions simultaneously and to ensure adequate sampling of backbone degrees of freedom (see Materials and methods and *Figure 1—figure supplement 2*). Then, we generated and analyzed ~30,000 models of the full-length *E. coli* Hfq hexamer. In the lowest energy (1%) subset of models, the acidic CTD residues (D97, E99, E100, and E102) frequently interact with basic residues on the rim (R16, R17, R19 and K47) and in the NTD (K3) (*Figure 1B*, top). By contrast, K31 on the distal face is not predicted to be contacted by the CTD, although K31 is highly accessible. This bias accords with prior observations that the CTD does not displace RNA from Hfq's distal face (*Santiago-Frangos et al., 2016*). As anticipated for a disordered domain, no single conformation dominated the ensemble of models (*Figure 1—figure supplement 3*). Rather, the acidic CTD tip was found to bind to various combinations of residues in the basic patch (*Figure 1C*, inset).

To confirm we were not simply observing the non-specific collapse of the disordered CTD onto the core, or enriching interactions between highly solvent-accessible polar residues, we repeated our simulations using a mutant Hfq in which the acidic CTD residues were replaced with polar or basic side chains (D97R-E99N-E100K-E102N). These mutations drastically decreased the frequency of predicted interactions between the basic core residues and CTD residues at positions 97, 99, 100 and 102 in our simulations (*Figure 1B*, bottom), without increasing predicted interactions between this mutant CTD and solvent-accessible acidic residues on the Hfq core (D9, E18, E37 and D40).

## Acidic CTD specifically binds Hfq rim

To determine whether the CTD interacts with the rim as predicted by our models, we used fluorescence anisotropy to measure the affinity of core Hfq (Hfq65) for a fluorescently-labeled CTD peptide, CTD-FITC (*Figure 2A* and *Figure 2—figure supplement 1*). CTD-FITC lacks residues 65–72 to avoid contributions to binding from this region, which packs against the Sm domain as one strand of the β-sheet (*Arluison et al., 2004*). Hfq65 bound to CTD-FITC with a $K_d$ of 2.9 μM Hfq monomer in low salt buffer (cyan in *Figure 2B*) and 22 μM in a higher salt buffer (*Figure 2—figure supplement 2*). These interaction strengths are meaningful even at higher ionic strength, because the effective concentration of each individual acidic CTD tip is roughly 350 μM in the full-length protein (see Materials and methods).

Binding of the CTD-FITC peptide to Hfq65 was weakened by mutations in the basic rim residues R16A, R19D and K47A (*Figure 2B*), which frequently interact with the CTD in our computational models (*Figure 1B*). Although we were not able to test the predicted interactions between NTD K3 and the CTD (see Materials and methods), K3 is also known to form electrostatic interactions with the RNA backbone (*Dimastrogiovanni et al., 2014*). In contrast, mutation of a surface-accessible polar residue (Q35A) close to the binding interface (*Figure 1C*, inset), slightly enhanced CTD binding (*Figure 2B*). Intriguingly, A35 is common in Hfq from γ-proteobacteria. Finally, a CTD peptide containing the mutated C-terminal tip (RSNKTN) was not able to bind Hfq65, confirming that the acidic residues on the CTD peptide are necessary for this interaction (grey in *Figure 2B*).

## CTD-bound core residues play a role in RNA annealing

To determine how much core residues that bind the CTD contribute to Hfq's RNA annealing activity, we compared the effect of rim mutations on the rate of base pairing between an RNA molecular beacon and the 16 nt Target RNA by stopped-flow fluorescence spectroscopy (*Figure 2C*) (*Hopkins et al., 2011*). In the absence of competition from the CTD, the rate of annealing in this

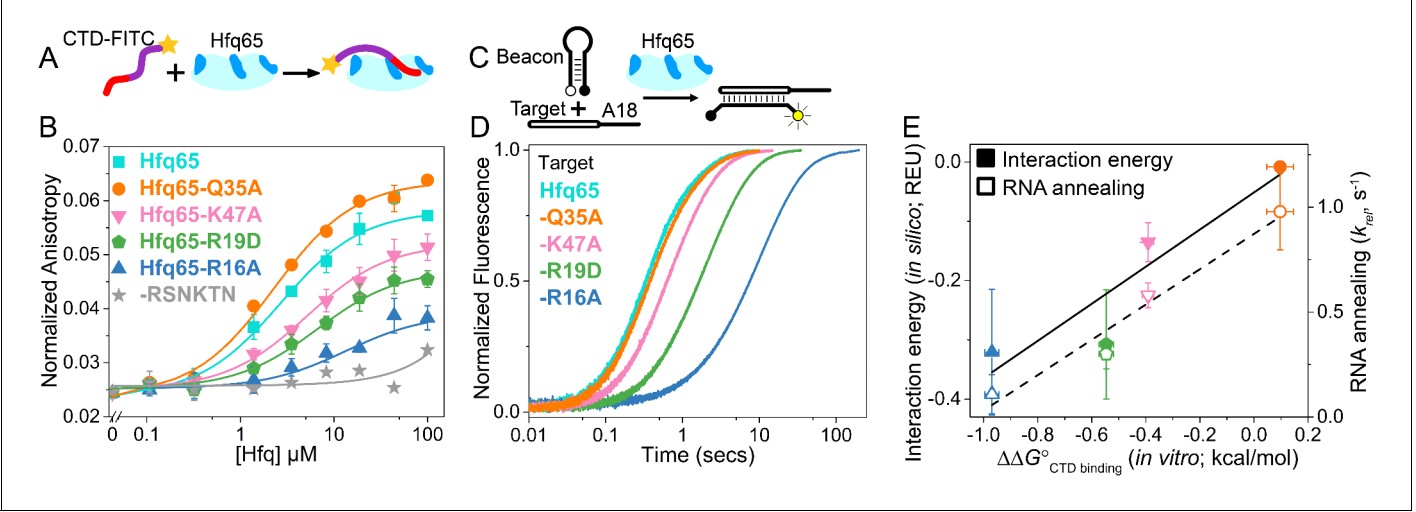

**Figure 2.** CTD and RNA occupy the same rim binding site. (**A**) Scheme for in vitro binding of fluorescent CTD-FITC peptide by Hfq core. The CTD linker is shown in purple, the acidic tip in red and the N-terminal FITC as a yellow star. Hfq core is shown in cyan, with basic rim patches in dark blue. (**B**) Binding of CTD-FITC to variants of Hfq65 core at 30 °C. 45 nM CTD-FITC was titrated with 0–100 µM Hfq monomer in duplicate, and the average (±s.d.) was fit to **Equation 5** (Materials and methods). (**C**) Reaction scheme for annealing an RNA molecular beacon to a target RNA (open bar) (**Hopkins et al., 2011**; **Panja et al., 2015**). (**D**) Progress curves for annealing 50 nM molecular beacon and 100 nM Target by 50 nM Hfq65 hexamer at 30°C, measured by stopped-flow fluorescence. See **Figure 2—figure supplement 3** for further data. (**E**) Contribution of core residues to CTD binding. Interaction energy (Expected Energetic Contribution; EEC) in silico for a core residue in the Rosetta models (solid symbols and solid line; adjusted $R^2 = 0.77$) or the average annealing rates for Target and Target-A18 relative to Hfq65 (open symbols and dashed line; adjusted $R^2 = 0.94$) versus experimental CTD binding energy (ΔΔG°) for each Hfq65 variant. The binding energy, $\Delta\Delta G° = -RT\ln(K_d^{MUT}/K_d)$, reflects the perturbation to CTD binding by a mutation in Hfq65. The interaction energy in silico or EEC is defined as the average energy of a tail–core interaction multiplied by the average number of tail–core interactions per model (**Figure 1B**, top and **Equation 3**). The relative annealing rate for Hfq65 variants, $k_{rel} = k_{obs}^{MUT}/k_{obs}^{WT}$, is <1 if the mutated residue is important for RNA annealing.

DOI: https://doi.org/10.7554/eLife.27049.006

The following figure supplements are available for figure 2:

**Figure supplement 1.** SDS-PAGE of purified Hfq variants.

DOI: https://doi.org/10.7554/eLife.27049.007

**Figure supplement 2.** Hfq65 binding to CTD-FITC in a higher salt buffer.

DOI: https://doi.org/10.7554/eLife.27049.008

**Figure supplement 3.** RNA annealing kinetics by Hfq65 and variants.

DOI: https://doi.org/10.7554/eLife.27049.009

assay depends only on interactions between the two RNAs and the Hfq core. As previously observed (**Santiago-Frangos et al., 2016**), Hfq65 is highly active in single-turnover annealing assays (**Figure 2—figure supplement 3A**). The observed annealing rate was most diminished by the loss of basic residues, especially the conserved R16A, and relatively unaffected by the mutation Q35A (**Figure 2D**). Similar results were obtained with Target-A18, which anchors to the distal face (**Figure 2—figure supplement 3B**). The average relative annealing rates of Hfq65 variants correlated well with the importance of each residue for CTD binding in vitro (**Figure 2E**), suggesting that the CTD peptide and the RNA interact with the same residues on the rim of Hfq.

The predictive value of our computational approach was validated by a direct correlation between the experimentally measured contribution (ΔΔG°) of each core residue for CTD binding with the predicted Expected Energetic Contribution (EEC) of that core residue to interactions with the acidic CTD in silico (solid symbols and solid line, **Figure 2E**). EEC is defined as the average energy of a tail–core interaction multiplied by the average number of tail–core interactions per model. The absolute binding and simulated interaction energies cannot be directly compared because the peptide binding assay is performed in trans rather than in cis, and the Rosetta Energy does not account for entropic contributions to binding. Nevertheless, amino acids that most strongly impacted the free energy of CTD binding when mutated, also had larger contributions to CTD binding in silico (solid symbols and solid line, **Figure 2E**; linear regression p-value=0.078), and had

stronger effects on Hfq65 RNA annealing activity *in vitro* (open symbols and dashed line, *Figure 2E*; linear regression p-value=0.020).

## Nucleic acids compete with the CTD for binding the Hfq core

If the CTD peptide and RNA interact with the same basic Hfq residues, direct competition between the two could explain how the CTD triggers the release of annealed RNAs from Hfq (*Santiago-Frangos et al., 2016*) and why it increases the stringency of RNA or DNA binding. To examine whether nucleic acids are in competition against the CTD for binding to the Hfq core, we compared the ability of different nucleic acids to displace CTD-FITC from a preformed Hfq65·CTD-FITC complex (*Figure 3A*). The strength of competition was expressed as the concentration of nucleic acid needed to displace 50% of CTD-FITC from Hfq65, $IC_{50}$.

Natural sRNAs strongly competed against the CTD-FITC peptide, in keeping with their strong (~10 nM) affinity for Hfq (*Figure 3B*). Short RNA and DNA oligomers that bind Hfq weakly were poorer competitors than natural sRNAs, as expected. In general, competition against the CTD peptide correlated with nucleic acid length, suggesting little sequence specificity, or that longer nucleic acids may occupy more than one basic patch (*Figure 3C*). However, minRCRB RNA, DNA1 and dsDNA2 deviated from this linear trend (dashed circles; *Figure 3C*). minRCRB, a stronger than expected competitor, consists of a stem-loop with a 5'-overhang, and has been shown to specifically bind to the rim (*Santiago-Frangos et al., 2016*; *Dimastrogiovanni et al., 2014*). Similarly, DNA1 possesses a stable minRCRB-like motif at its 5-end and was also a stronger competitor than expected based on its length. By contrast, the completely double-stranded dsDNA2 was a weaker competitor than expected, consistent with the CTD's ability to displace annealed dsRNAs from the rim of Hfq (*Santiago-Frangos et al., 2016*).

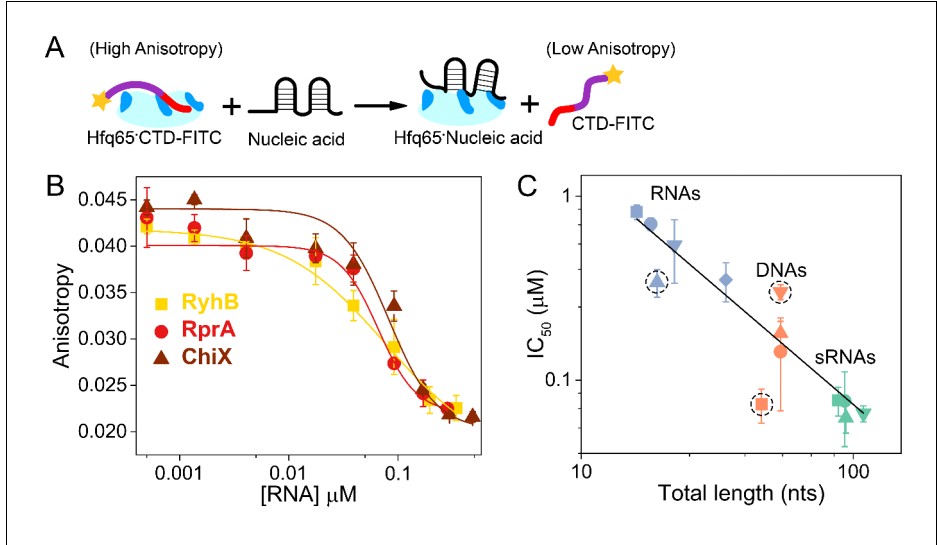

**Figure 3.** Nucleic acids displace the CTD from the Hfq core. (**A**) Scheme for *in vitro* competition of the CTD-FITC·Hfq65 interaction by nucleic acids. (**B**) Example titrations of the CTD-FITC·Hfq65 complex with sRNA. Titrations were done in duplicate, and the averages (±s.d.) fit to *Equation 6* (Materials and methods) to determine the $IC_{50}$.(**C**) $IC_{50}$ values for nucleic acids of different lengths (blue, RNA oligomers; orange, DNA oligomers; green, sRNAs). From shortest to longest: Target, A18, minRCRB, Target-U6, Target-A18, DNA1, DNA2, DNA2c, dsDNA2, ChiX, RyhB, DsrA, RprA). Dashed circles indicate nucleic acids that deviate from the linear relationship between log $IC_{50}$ and length (adjusted $R^2$ = 0.85).
DOI: https://doi.org/10.7554/eLife.27049.010

## Higher local concentration of acidic residues increases CTD autoinhibition

The results above indicate that an interaction between the acidic CTD tip and the basic rim inhibits RNA binding to the basic patch and stimulates release of dsRNA. If this model is correct, shortening the CTD should increase the local concentration of the acidic CTD tip and exacerbate autoinhibition. Alternatively, if the CTD acts as a polymer brush, a shorter CTD should exhibit less autoinhibition because it will exclude less volume around the Hfq core. To test these predictions, we generated the mutant Hfq-sCTD, which lacks residues 86–96 (inclusive). Truncation of this non-conserved 'linker' region is predicted to increase the local concentration of the acidic tip around the Hfq core roughly three-fold, from roughly 350 μM per CTD in Hfq102, to ~1220 μM per CTD in Hfq-sCTD (*Equation 4* in Materials and methods).

As previously shown (*Santiago-Frangos et al., 2016*), Hfq65, which lacks the CTD entirely, anneals the 16 nt Target and Target-U6 RNA about five times faster than full-length Hfq102 (100–60-fold vs. 20–10-fold; cyan and black in *Figure 4A*). In our model, this is because the CTD sweeps RNAs from the rim and proximal face of Hfq. By contrast, both proteins accelerate Target-A18 annealing roughly 100-fold compared to no Hfq, because this RNA remains anchored to the distal face of Hfq and resists CTD displacement (*Santiago-Frangos et al., 2016*). Importantly, the short-ened CTD linker (Hfq-sCTD) decreased annealing rates relative to Hfq102 for all RNA targets (*Figure 4A* and *Figure 4—figure supplement 1*), suggesting that access to the basic patch was more restricted.

Next, we used steady-state fluorescence anisotropy to examine how a shorter CTD affects the release of annealed dsRNA from the Hfq core (*Figure 4B*). Binding of Hfq102, Hfq65 or Hfq-sCTD to FAM-labeled D16 RNA increased the anisotropy of FAM fluorescence, as expected (*Figure 4D*). The smaller anisotropy of the Hfq65·D16-FAM complex is due to its smaller hydrodynamic drag, since all Hfqs have similar affinities for D16-FAM (*Table 1*). When complementary R16 RNA was added to the Hfq102·D16-FAM complex, the anisotropy decreased since most of the annealed dsRNA dissociated from Hfq102 (*Santiago-Frangos et al., 2016*) (black, *Figure 4B*). Whereas, when complementary RNA was added to the Hfq65·D16-FAM complex, the anisotropy increased (*Santiago-Frangos et al., 2016*), because a large proportion of the dsRNA remained bound to Hfq65 (cyan, *Figure 4C*). When the same experiment was done for Hfq-sCTD, the anisotropy decreased even fur-ther than for Hfq102, suggesting that more dsRNA was released when the CTD is shortened. This smaller anisotropy cannot be explained by the slightly smaller molecular weight of Hfq-sCTD, since the maximum anisotropies of Hfq102 and Hfq-sCTD ternary complexes during equilibrium binding experiments were very similar.

As a control, we also measured the relative affinities of each Hfq for the dsRNA product (D16-FAM·R16) versus ssRNA substrate (D16-FAM), $K_{rel} = K_d(P)/K_d(S)$ (*Table 1*). The $K_{rel}$ of each protein corresponded to the efficiency of product release in the anisotropy experiment (*Figure 4C* and *Table 3*): Hfq102 had a high $K_{rel} = 7.5$, Hfq65 had a low $K_{rel} = 2.3$, and Hfq-sCTD had the highest $K_{rel} = 14.0$. Thus, the number of non-conserved residues between the acidic tip and the core of Hfq dictates the stringency of CTD autoinhibition and the efficiency of dsRNA displacement, presumably by controlling the effective concentration of the acidic tip around the Hfq core.

## CTD limits sRNA-mRNA association

To determine whether the above results on RNA oligomers apply to natural RNA substrates for Hfq, we examined the annealing of the Class II sRNA ChiX to the mRNA *chiP* via electrophoretic mobility shift assays (EMSAs). Because the mRNA targets of class II sRNAs interact with the rim of Hfq (*Schu et al., 2015*; *Małecka et al., 2015*), we reasoned that this sRNA-mRNA pair would be sensitive to displacement by the CTD. Low nanomolar amounts of ChiX and *chiP* anneal very slowly at 10°C, in the absence of Hfq (*Figure 4—figure supplement 2*). Whereas, Hfq102, Hfq65 and Hfq-sCTD all form a ternary complex with ChiX and *chiP* within 20 s (*Figure 4D*), reaching equilibrium in a few minutes (*Figure 4—figure supplement 2*). These results demonstrate that the CTD is not necessarily required for annealing longer natural RNAs. In addition, Hfq65 formed the most stable ChiX·Hfq·*chiP* ternary complex, with almost no ChiX-Hfq65 binary complex remaining after 20 s (*Figure 4E*). Less ternary complex was formed by Hfq102, and the least by Hfq-sCTD. These results are consistent with the idea that the CTD limits access of *chiP* mRNA to the rim of Hfq.

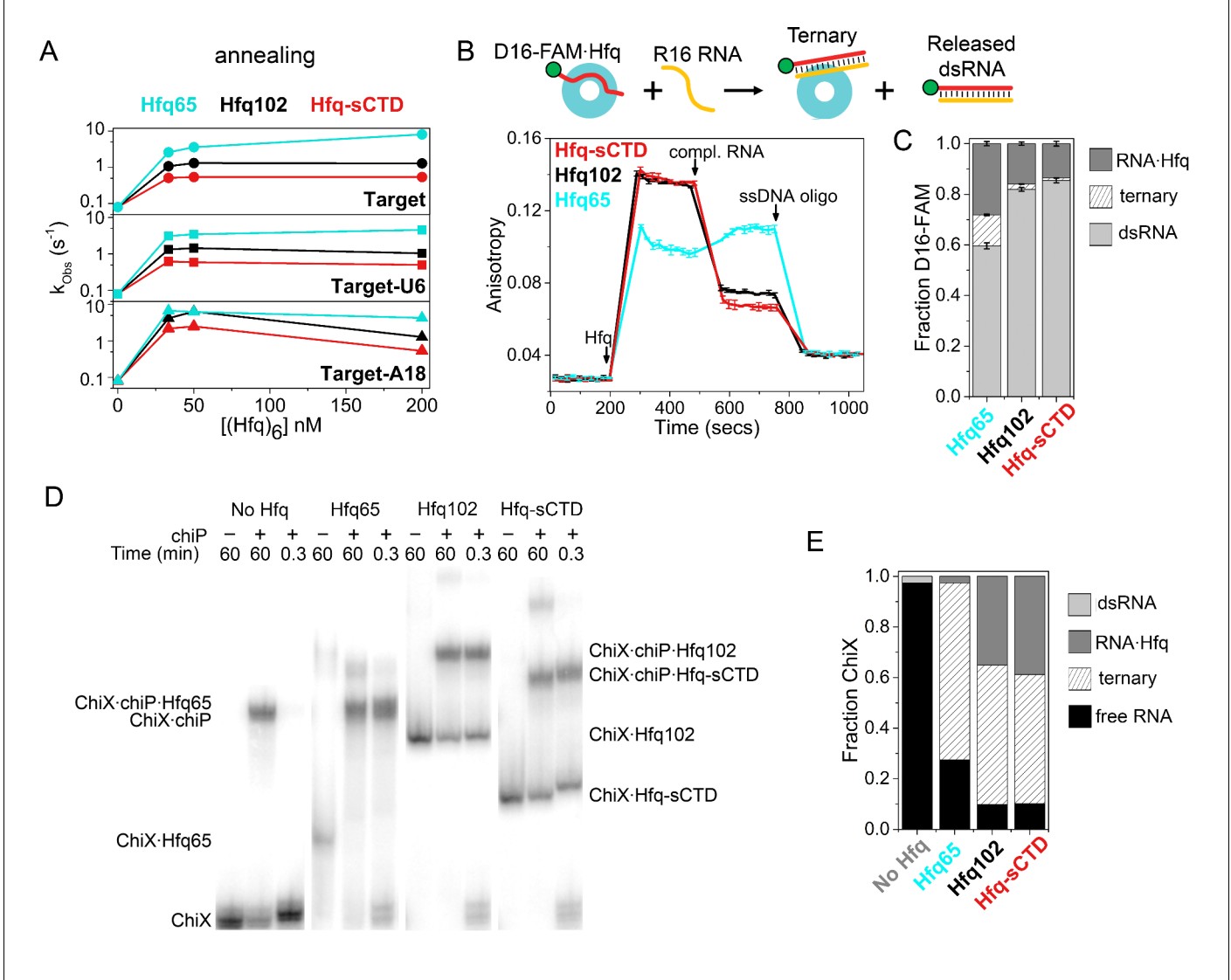

**Figure 4.** Shortening the CTD increases auto-inhibition by acidic residues. (**A**) Observed annealing rate constants for 100 nM target RNA and 50 nM molecular beacon with 0–200 nM Hfq hexamer at 30°C, measured by stopped-flow fluorescence. Black, Hfq102; red, Hfq-sCTD; cyan, Hfq65. (**B**) Fluorescence anisotropy assay for RNA binding and release. In stopped-flow FRET experiments, the dsRNA is released after pairing with its complementary strand (*Panja et al., 2013*; *Santiago-Frangos et al., 2016*). D16-FAM RNA (50 nM) was allowed to bind 50 nM Hfq hexamer. The Hfq·D16-FAM complex was challenged with 50 nM complementary R16 RNA. Most of the D16-FAM·R16 product is released from Hfq102 and Hfq102-sCTD, but not Hfq65. Remaining RNA was displaced from Hfq by excess ssDNA (400 nM DNA2). The averages and standard deviations for three trials are plotted for each Hfq variant. (**C**) Molar fractions of D16-FAM·R16 product released (light grey), remaining D16-FAM·Hfq·R16 ternary complex (hatched), and Hfq·D16-FAM binary complex (dark grey) calculated from *Equation 7* (Materials and methods), based on the anisotropies at the end of the annealing phase and the maximum anisotropies of ternary complexes from equilibrium binding experiments (*Table 3*). (**D**) Annealing of 5 nM $^{32}$P-ChiX sRNA with 30 nM *chiP* mRNA without Hfq, or with Hfq65, Hfq102 or Hfq-sCTD, as indicated above each lane. Samples were loaded on a native polyacrylamide gel 60 min or 20 s after the components were mixed. Full gel images are in *Figure 4—figure supplement 2*. (**E**) Fractions of free ChiX (black), ChiX·*chiP* dsRNA (light grey), ChiX·Hfq·*chiP* ternary complex (hatched) and ChiX·Hfq binary complex (dark grey) after 4 min of annealing, as analyzed by EMSA (*Figure 4—figure supplement 2*).

DOI: https://doi.org/10.7554/eLife.27049.011

The following figure supplements are available for figure 4:

**Figure supplement 1.** Effect of CTD length on RNA oligomer annealing kinetics.
DOI: https://doi.org/10.7554/eLife.27049.012

**Figure supplement 2.** Effect of CTD length on sRNA·mRNA annealing.
DOI: https://doi.org/10.7554/eLife.27049.013

**Table 1.** Equilibrium dissociation constants for Hfq.
Values are the mean ± SD of three independent experiments. $K_d$ values were determined by fluorescence anisotropy (see Methods).
*Values were previously determined (*Santiago-Frangos et al., 2016*).

| | $K_d$ (nM hexamer) | | | Hill coefficient | | |
| --- | --- | --- | --- | --- | --- | --- |
| RNA | $Hfq_{102}$ | $Hfq_{65}$ | Hfq-sCTD | $Hfq_{102}$ | $Hfq_{65}$ | Hfq-sCTD |
| D16-FAM | 15.5 ± 0.9* | 20.0 ± 1.3* | 12.9 ± 2.1 | 0.7 ± 0.1* | 0.6 ± 0.1* | 0.8 ± 0.1 |
| D16-FAM·R16 | 117 ± 12 | 45.5 ± 2.4 | 181 ± 12 | 1.2 ± 0.1 | 1.0 ± 0.1 | 1.0 ± 0.1 |
| minRCRB | 13.9 ± 1.5* | 6.46 ± 0.7* | 20.1 ± 1.9 | 1.4 ± 0.2* | 2.5 ± 0.6* | 1.1 ± 0.1 |

DOI: https://doi.org/10.7554/eLife.27049.014

## DNA binding is regulated by the CTD

Hfq binds dsDNA and has been reported to associate with the bacterial chromosome (*Kajitani et al., 1994*; *Takada et al., 1997*; *Azam and Ishihama, 1999*; *Updegrove et al., 2010*; *Jiang et al., 2015*). Since the CTD peptide competes with DNA for binding to the Hfq core (*Figure 3B*) and inhibits binding of dsRNA and dsDNA more strongly than ssRNA (*Table 1* and *Figure 3*), we asked whether the CTD modulates binding of Hfq to DNA. A change in DNA binding could alter the distribution of Hfq within the cell.

We quantified Hfq102 binding to linearized pUC19 DNA by measuring the change in DNA electrophoretic mobility in 1.5% agarose (*Figure 5A,B*). The mobility of the DNA-Hfq complexes decreased with added Hfq102, indicating increasing numbers of Hfq hexamers bound per DNA. The complexes exhibited uniform mobility at each protein concentration, however, except for a smear below the main band that may arise from dissociation of Hfq during electrophoresis (*Figure 5C*, top). This pattern is consistent with an equal (non-cooperative) distribution of Hfq102 between DNA molecules, although Hfq was previously suggested to bind DNA cooperatively (*Cech et al., 2016*). Cooperative binding to neighboring sites on the DNA would result in distinct bound and unbound populations of pUC19 (*Tapias et al., 2000*; *Kozlov et al., 2010*), which we do not observe here. The rim mutation R16A lowered the maximum gel retardation and increased the intensity of the 'smear', consistent with a reduced affinity of this mutant for dsDNA (*Updegrove et al., 2010*).

In contrast to the results with the full-length protein, we observed two behaviors when Hfq65 core interacted with pUC19 DNA. At low protein concentrations, the DNA was sparsely bound by Hfq65, resulting in a small mobility shift (*Figure 5B*). At higher protein concentrations, the DNA formed aggregates that were too large to enter the gel, resulting in a loss of signal (*Figure 5B* and *Figure 5D*, top). Aggregation of the DNA was confirmed by solubilization with Proteinase K or by pelleting assays (*Figure 5—figure supplement 1*). The rim mutation Hfq65-R16A rescued the formation of insoluble aggregates, confirming that dsDNA interacts with the basic rim and not the CTD itself, as previously proposed (*Jiang et al., 2015*). Moreover, the Hfq65-R16A complexes migrated more slowly than those formed by similar concentrations of Hfq102 (*Figure 5C* and *Figure 5D*, bottom), suggesting that Hfq65 and Hfq65-R16A bind pUC19 at higher densities than Hfq102. Thus, the CTD appears to limit Hfq102 binding to DNA, perhaps by maintaining a regular spacing between Hfq hexamers or by enforcing a dynamic equilibrium between bound and free protein. By contrast, when the Hfq core is exposed by deletion of the CTD, Hfq65 binds and aggregates dsDNA indiscriminately.

To examine which Hfq surfaces bind dsDNA, we challenged complexes of 0.5 µM Hfq102 hexamer and pUC19 DNA with 0–2 µM RNA or DNA oligomers that interact with different sites on Hfq (*Figure 5E* and *Figure 5—figure supplement 2*). A18 RNA that binds the distal face of Hfq did not compete for DNA binding (*Figure 5E*, top). However, when the 16 nt Target RNA that weakly interacts with the rim is appended to A18 (Target-A18), the oligomer strongly competed against DNA for binding to Hfq (*Figure 5E*). Target-U6 was also a good competitor, whereas the 16 nt Target sequence alone, which binds Hfq102 weakly at the rim (1 µM), was a poor competitor. DNA1 and minRCRB, which bind to the rim of Hfq (*Santiago-Frangos et al., 2016*; *Dimastrogiovanni et al., 2014*) (*Figure 3C*), but have low affinities for Hfq102, were weak competitors for DNA (*Figure 5F*).

Finally, the sRNAs ChiX and RyhB were the strongest competitors, with competition saturating at a 1:1 ratio of sRNA to Hfq102 hexamer (*Figure 5F*).

Overall these data indicate that the CTD does not directly bind DNA as previously suggested (*Updegrove et al., 2010*; *Jiang et al., 2015*), but rather modulates the ability of the Hfq core to bind DNA, so that the extent of binding is limited and Hfq-DNA complexes remain soluble. Our data do not conflict with a low-resolution SANS model that suggests Hfq binds perpendicularly to the DNA duplex with a slight tilt (*Jiang et al., 2015*), but suggest that this occurs when the basic rim of Hfq interacts with the phosphate backbone of the DNA duplex. We note that the potent competition from sRNAs, which are more numerous than Hfq in the cell (*Wagner, 2013*), calls into question the hypothesis that *E. coli* Hfq regulates cellular processes via DNA binding (*Sobrero and Valverde, 2012*; *Cech et al., 2016*).

### CTD–core interactions in other bacterial Hfq's

Our results on *E. coli* Hfq show that the strength and frequency of CTD–core interactions depend on the number of basic residues in the core, the acidic residues in the CTD, and the linker length. Thus, our proposed mechanism for CTD-core interactions can be used to predict how the degree of CTD autoinhibition may vary among bacterial Hfq's. We applied our *de novo* modeling procedure to estimate the CTD–core interactions in four other bacterial Hfqs (*Figure 6A*) for which the genetic function and *in vitro* annealing activity have been previously characterized (*Zheng et al., 2016*; *Bohn et al., 2007*; *Liu et al., 2010*; *Rochat et al., 2015*; *Christiansen et al., 2004*; *Oglesby-Sherrouse and Vasil, 2010*). We examined low energy models of each Hfq hexamer, and compared how frequently acidic CTD residues interact with basic rim and NTD residues ('on-target') versus other core residues ('off-target') (see Materials and methods) (*Figure 6B,C*). This comparison was quantitatively expressed as the difference in the EEC of on-target and off-target interactions (ΔEEC).

For *E. coli* Hfq, an active chaperone with a basic rim patch and long CTD, the CTD tip tended to interact with basic residues on the rim and NTD more often and more strongly than with other residues, resulting in ΔEEC = −1.11 ± 0.20 REU. This was also true for *Listeria monocytogenes* Hfq (ΔEEC = −1.08 ± 0.22 REU). In contrast, *Bacillus subtilis* (ΔEEC = 0.51 ± 0.14 REU) and *Staphylococcus aureus* (ΔEEC = 0.19 ± 0.05 REU) Hfq, which are inactive in our *in vitro* annealing assay (*Zheng et al., 2016*), did not exhibit specific CTD-core interactions. Finally, in models of full-length *Pseudomonas aeruginosa* Hfq, the CTD adopts an extended β conformation that wraps over the rim of the hexamer and places the C-terminal acidic residues near the weakly basic NTDs (ΔEEC = −0.30 ± 0.04 REU) (*Figure 6C*). In the absence of the NTD, however, the CTDs dock with R16 and K17 on the rim (ΔEEC = −0.50 ± 0.18 REU). Thus, Hfqs that do not anneal RNA *in vitro* tend to possess shorter, less acidic CTDs that form weaker and less frequent interactions with the basic rim and NTD *in silico* (*Figure 6D*). There is a similar trend between ΔEEC and the importance of Hfq for sRNA regulation in each bacterium (*Zhang et al., 2013*; *Bohn et al., 2007*; *Liu et al., 2010*; *Rochat et al., 2015*; *Oglesby-Sherrouse and Vasil, 2010*; *Tsui et al., 1994*; *Nielsen et al., 2010*; *Rochat et al., 2012*).

In the above examples, both the CTD and the core co-vary between different species. We next asked whether the CTD conferred specificity or strength to CTD-core interactions. We modeled an Hfq chimera consisting of the highly basic *E. coli* Sm core, fused to the shorter and slightly less acidic *B. subtilis* CTD. In our models, the *B. subtilis* CTD contacted K3 in the NTD and R17 on the rim more frequently than *E. coli* CTD (*Figure 6—figure supplement 1A*), but contacted R16 and R19, which are functionally very important (*Figure 1* and *Figure 2*), less frequently than *E. coli* CTD (*Figure 6—figure supplement 1A*). This was corroborated with fluorescence anisotropy results showing that *E. coli* Hfq65 binds a BsCTD-FITC peptide about three times more weakly than its own CTD (8.7 μM vs. 2.9 μM; *Figure 6—figure supplement 1B*). Although we have shown that a foreign CTD can bind the core of *E. coli* Hfq, the 'specificity' of this interaction may have been lost.

## Discussion

We previously found that the flexible CTD of *E. coli* Hfq sweeps RNAs from the proximal and rim surfaces of the Hfq ring by an unknown mechanism (*Santiago-Frangos et al., 2016*). Because the mechanism was not known, it was not possible to predict whether other bacterial Hfq CTDs, which are highly variable in sequence composition and length, would perform similar functions. Here, we have

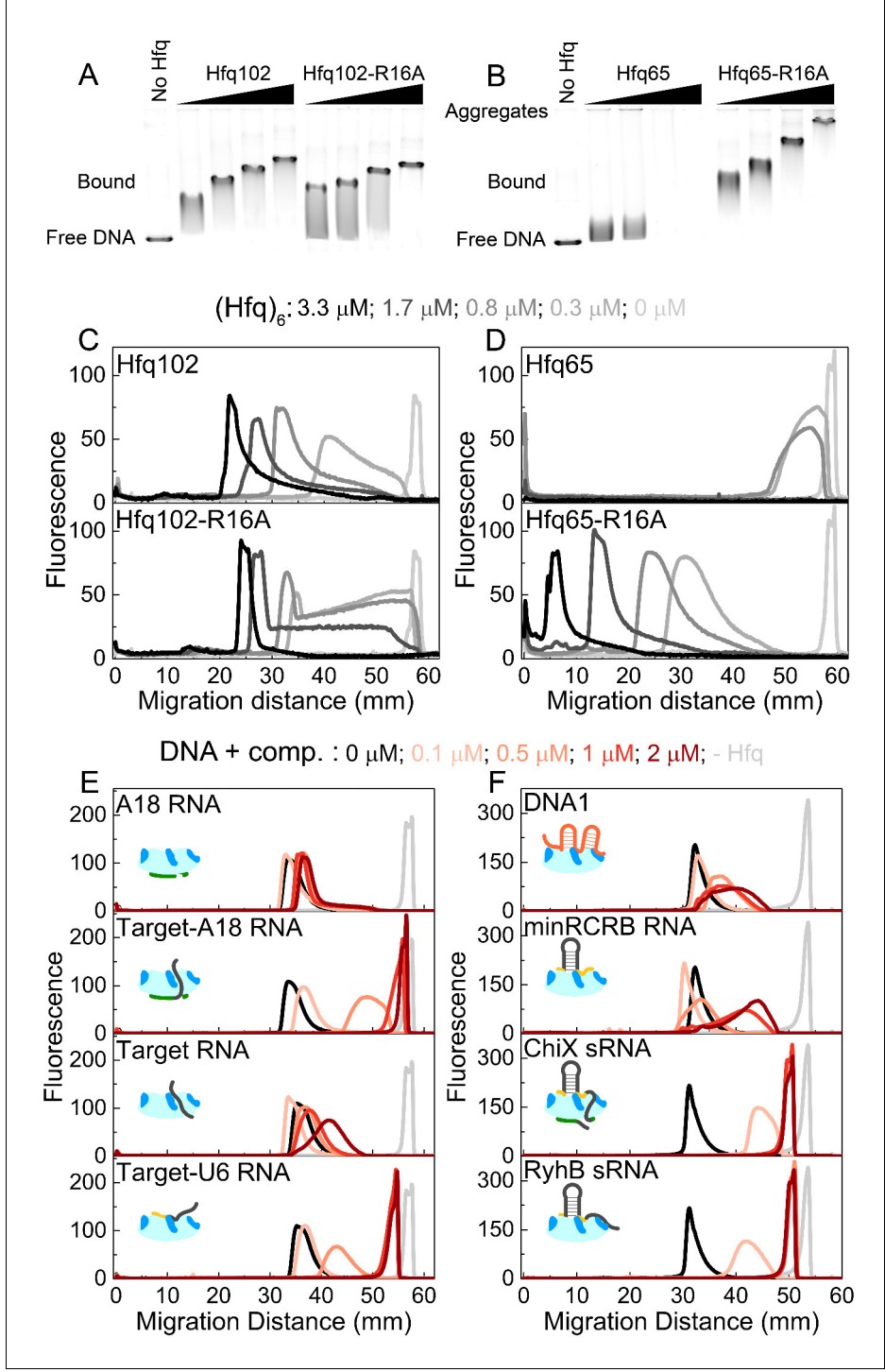

**Figure 5.** CTD and RNAs inhibit DNA binding to Hfq by obstructing the basic rim patch. (**A, B**) Agarose gel electrophoretic mobility shift assays of 0–3.3 µM hexamer Hfq102, Hfq102-R16A, Hfq65 or Hfq65-R16A binding to 6 nM linear pUC19 DNA (2635 nts) at 25°C, stained with SYBR Gold. Hfq65 forms large aggregates that fail to enter the gel (**Figure 5—figure supplement 1**). (**C, D**) Line densitometry of DNA migration in (**A, B**). Free pUC19 (no Hfq) is shown in light grey. Samples with increasing Hfq concentration are shown in darker shades of grey. (**E, F**) RNA competition. Complexes of 0.5 µM Hfq102 hexamer and 6 nM linear pUC19 (black) were challenged by 0–2 µM RNA or DNA competitor (darker shades of red). pUC19 control in the absence of Hfq is shown in light grey. Icons indicate the binding surface for each competitor. See **Figure 5—figure supplement 2** for gel images.

DOI: https://doi.org/10.7554/eLife.27049.015

*Figure 5 continued on next page*

*Figure 5 continued*

The following figure supplements are available for figure 5:

**Figure supplement 1.** Hfq CTD prevents aggregation with DNA.

DOI: https://doi.org/10.7554/eLife.27049.016

**Figure supplement 2.** Competition of Hfq-DNA interactions by small RNAs.

DOI: https://doi.org/10.7554/eLife.27049.017

used computational models and experiments to show that the acidic tip of the CTD directly displaces RNA from basic patches on the rim of Hfq. The CTD's mimicry of nucleic acids is supported by direct competition between nucleic acids and the CTD for binding the Hfq core, and stronger auto-inhibition when the linker connecting the acidic tip to the core is shortened (Hfq-sCTD). The good agreement between the modeled CTD–core contacts and the contributions of individual residues to the measured CTD binding energies and to RNA annealing validates our modeling approach, and further suggests that nucleic acids and the acidic tip of the CTD interact with the same residues in the Hfq core. As expected for a nucleic acid mimic, CTD·core interactions are dominated by electrostatics and exhibit a salt dependence (*Figure 2* and *Figure 2—figure supplement 2*) similar to that seen for the autoregulatory CTD of HTLV-1 nucleocapsid (*Qualley et al., 2010*).

Our results show that competition between the CTD and RNA improves the efficiency of *E. coli* Hfq's chaperone activity while increasing the stringency of substrate selection. In our model, sRNA and mRNA substrates are recruited through specific interactions with the proximal or distal face of the Sm ring. When complementary RNA segments engage one or more basic patches on the rim of Hfq, these interactions favor nucleation and zippering of the double helix (*Panja et al., 2013*; *Panja et al., 2015*). Transient interactions between the CTD and the rim leads to the displacement of the dsRNA product, preventing strand dissociation and recycling Hfq. In support of the model, we observe that Hfq102 binds dsRNA less strongly than single-stranded RNA, whereas Hfq65 binds them more similarly (*Table 1*).

We propose that the CTD makes Hfq a more selective RNA binding protein by inhibiting access to its rim. Single-stranded nucleic acids compete with the CTD approximately in proportion to length, suggesting that nucleic acid binding to the Hfq rim has low sequence-specificity. Consequently, short RNAs that only bind the rim, such as the 16 nt Target, weakly compete with the CTD and are poor substrates for annealing. By contrast, sRNAs or mRNAs that specifically bind the proximal or distal face of Hfq strongly compete against the CTD, and gain access to the basic patch on the rim. We previously showed that the CTD increases competition among *E. coli* sRNAs, resulting in different levels of sRNA accumulation in the cell (*Santiago-Frangos et al., 2016*). It remains to be shown whether the CTD also increases the stringency of target site selection. The six CTDs, which are disordered and mobile (*Beich-Frandsen et al., 2011b*), exclude a large volume around the core of *E. coli* Hfq (*Figure 1C*). This excluded volume is expected to limit the number of sRNAs that may bind *E. coli* Hfq at any one time, perhaps further increasing the stringency of RNA selection.

The hyper-variability of the Hfq CTD among different bacteria points to a continuous optimization of autoinhibition and binding selectivity, possibly in response to the acquisition or evolution of novel sRNA-mRNA regulatory pairs (*Peer and Margalit, 2014*). A balance of interactions at the rim of Hfq is needed, since a CTD that inhibits RNA binding too strongly may adversely affect interactions with genuine RNA substrates (*Figure 4*). Conversely, our DNA binding results show that an exposed basic patch can bind DNA (and RNA) indiscriminately, hinting that a basic patch necessitates co-evolution with a 'protective' CTD. Intriguingly, most Hfqs contain acidic sequences at the end of the CTD (*Figure 1—figure supplement 1*), despite a general bias toward basic residues at protein C-termini (*Berezovsky et al., 1999*) and within intrinsically disordered domains (*Williams et al., 2001*; *Lise and Jones, 2005*). Additionally, the CTDs of *E. coli* Hfq are long enough to contact the rims of neighboring monomers (*Figure 6—figure supplements 2–3*), which may explain the contribution of the CTD to hexamer stability (*Vincent et al., 2012*), and inter-hexamer contacts (*Figure 2—figure supplement 1*).

Computational modeling provided atomic-scale insight to the accessible conformations of the disordered N- and C- termini of *E. coli* Hfq (*Figure 1C*). Our experimentally validated EEC metric

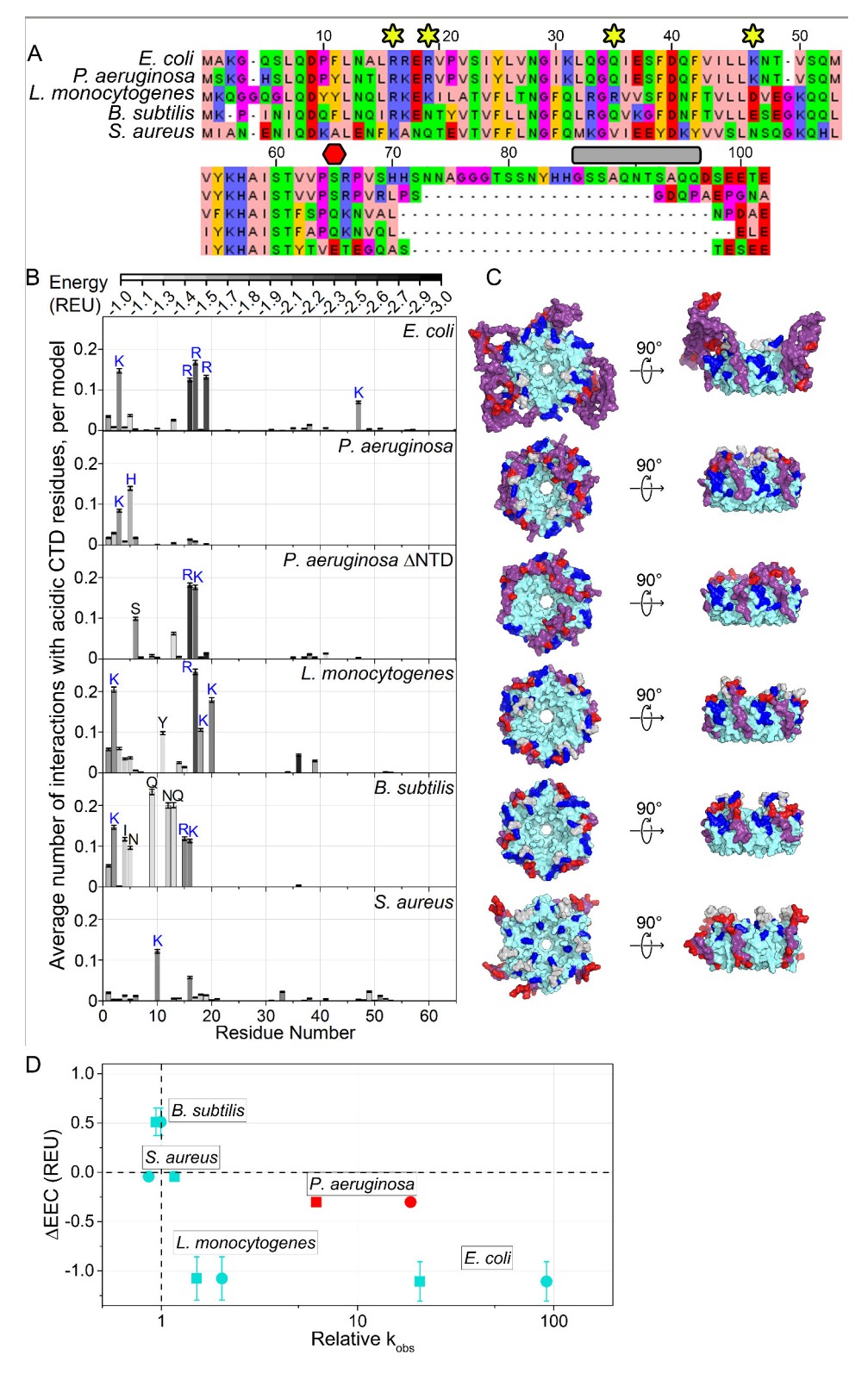

**Figure 6.** Acidic CTD and basic patch correlate with the chaperone activity of bacterial Hfq's. (**A**) Alignment of modeled Hfq sequences in order of decreasing *in vitro* RNA annealing activity. Residues are numbered according to the *E. coli* sequence. Yellow stars, residues mutated in this study; red hexagon, last residue in Hfq65; grey box, linker removed in Hfq-sCTD. (**B**) Average number of favorable interactions per model for each core residue with at least one acidic CTD residue (*Equation 1*) in the lowest energy models (≤1%). As in *Figure 1B*. Number of residues with < 2.0 A² accessible

*Figure 6 continued on next page*

*Figure 6 continued*

surface area: *P. aeruginosa,* 10; *L. monocytogenes,* 25; *B. subtilis,* 19; *S. aureus,* 12. (C) Top-down (proximal face) and side (rim) views of example low-energy models for each Hfq, as in *Figure 1C*. (D) Relative RNA beacon annealing rate for Target-U6 (boxes) and Target-A18 (circles) in Hfq vs. no Hfq (relative $k_{obs}$) versus the specificity of predicted CTD–core interactions (ΔEEC) for *B. subtilis, S. aureus, L. monocytogenes* and *E. coli* Hfq (blue), and *P. aeruginosa* Hfq, which is more active *in vitro* than predicted by its ΔEEC (red). Annealing data are from *Zheng et al. (2016)*.
DOI: https://doi.org/10.7554/eLife.27049.018

The following figure supplements are available for figure 6:

**Figure supplement 1.** CTD-core interactions in a chimeric Hfq.
DOI: https://doi.org/10.7554/eLife.27049.019
**Figure supplement 2.** Monomer distances of CTD–core interaction.
DOI: https://doi.org/10.7554/eLife.27049.020
**Figure supplement 3.** Monomer distances for interactions between acidic CTD residues and basic core residues.
DOI: https://doi.org/10.7554/eLife.27049.021

(*Figure 2E*) defines residue–residue interactions more accurately with respect to experimental data than commonly used distance cutoffs (Cα–Cα and Cβ–Cβ) (*Kleiger et al., 2009*; *Fischer et al., 2006*). This *de novo* modeling strategy was able to identify frequent and specific CTD–rim interactions in *Lm* and *Pa* Hfq, which act in sRNA regulation and annealing (*Panja et al., 2013*; *Zheng et al., 2016*; *Bohn et al., 2007*; *Liu et al., 2010*; *Rochat et al., 2015*; *Rochat et al., 2012*), but not for *Bs* and *Sa* Hfq (*Figure 6D*), in agreement with *in vitro* experiments. This suggests that the FloppyTail algorithm could be generally useful for predicting the interactions of disordered regions with ordered domains.

Many RNA and DNA binding proteins contain disordered or flexible domains that have been implicated in cooperativity, autoinhibition and liquid phase separation (*Trudeau et al., 2013*; *Varadi et al., 2015*; *Järvelin et al., 2016*). Hfq is an example of an emerging paradigm of autoregulation of nucleic acid binding by nucleic acid mimic peptides. Other examples in which a disordered CTD autoinhibits RNA or DNA binding include HTLV-1 NC (*Qualley et al., 2010*), *E. coli* gyrase (*Tretter and Berger, 2012*), *E. coli* ssDNA binding protein (*Kozlov et al., 2010*) and mammalian high-mobility group B1 (*Watson et al., 2007*). Unlike HTLV-1 NC, which also remodels RNA, the Hfq CTD gives rise to dynamic cycling of bound RNAs needed to chaperone sRNA-mRNA interactions. Our modeling procedure could be utilized to screen disordered domains found in kinases, such as myosin light chain kinases and protein kinase C (*Kobe and Kemp, 1999*), and nucleic acid binding proteins from all kingdoms of life (*Trudeau et al., 2013*; *Adams, 2003*; *Ward et al., 2004*; *Wang et al., 2016*).

## Materials and methods

### Hfq alignments and sequence logos

All Hfq gene sequences were taken from Uniprot (*UniProt Consortium, 2015*). 5359 sequences were aligned using the G-INS-1 algorithm on MAFFT webservers (*Yamada et al., 2016*). This alignment was reduced using CD-HIT (*Li and Godzik, 2006*) and Max-Align (*Gouveia-Oliveira et al., 2007*). An unrooted, neighbor-joining tree of the remaining 985 non-redundant, representative, sequences was made on MAFFT webservers (*Yamada et al., 2016*). Sequence logos of re-aligned sequences from chosen clusters were generated using WebLogo (*Crooks et al., 2004*).

### Computational modeling of the intrinsically disordered regions
#### Structure preparation
The crystal structures of *E. coli* (1HK9) (*Sauter et al., 2003*), *P. aeruginosa* (1U1S) (*Nikulin et al., 2005*), *L. monocytogenes* (4NL2) (*Kovach et al., 2014*), *B. subtilis* (3HSB) (*Someya et al., 2012*), and *S. aureus* (1KQ1) (*Schumacher et al., 2002*) Hfqs were used as starting points for the computational modeling. All crystal structures contained the hexameric form of Hfq, except for 4NL2, for which we generated the biologically relevant hexamer using the reported symmetry operations. Missing

residues were appended or prepended to the crystal structures in the following manner. First, on a single subunit, absent N-terminal residues were prepended and all N-terminal residues predicted to be disordered (*Buchan et al., 2013*; *Jones and Cozzetto, 2015*) were initialized in an extended conformation, with backbone dihedral angles set to: $\phi = -135°$ and $\psi = 135°$. Since the Hfq hexamer is C6 symmetric, the modified subunit could be symmetrized to all other subunits. The same process was repeated to append C-terminal residues, except the base of the tail (residues 64–69 in 1HK9, 1U1S, and 1KQ1, 66–71 in 4NL2, and 62–67 in 3HSB) was 'kinked' to point proximally as in the 1HK9 structure. For the RSNKTN tail mutant, side chains were mutated using the PyMOL 'mutate' function. The structures with extended termini were 'relaxed' with constraints, using Rosetta (*Conway et al., 2014*), to eliminate energetically unfavorable atomic clashes, before modeling.

## Modeling

A modified version of the FloppyTail algorithm (*Kleiger et al., 2009*) was used to model the disordered termini (see Appendix 1 for step-by-step protocol). The source code is freely available to academic users through the RosettaCommons: www.rosettacommons.org. The FloppyTail algorithm generates hypothetical, low-energy conformations of disordered regions through two stages of modeling: (i) low-resolution modeling, where side chains are represented as single pseudo-atom centroids, with aggressive sampling of backbone conformational space and gradient-based minimization, and (ii) all-atom modeling, where all side-chain atoms are restored, with fine sampling of backbone conformational space, side-chain optimization, and minimization. We adapted the original algorithm to permit simultaneous modeling of multiple disordered termini and to more extensively sample conformation space. In our simulations, any Hfq region predicted to be disordered was allowed to move and approximately 500 backbone moves (changes in ϕ/ψ angles) were attempted per disordered residue (*Kleiger et al., 2009*) attempted ~25 backbone moves per disordered residue). Non-disordered residues had no backbone motion, but were permitted to sample side-chain conformations. In total, simulations were used to generate ~30,000 hypothetical structures for each species' Hfq.

## Analysis

PyRosetta (*Chaudhury et al., 2010*) was used to evaluate the energies of pairwise residue–residue interactions. Pairwise energies were computed with the *talaris2014* energy function (*O'Meara et al., 2015*), comprised of terms capturing van der Waals, solvation, hydrogen bonding and electrostatic interactions. If a pairwise energy was unfavorable (0 or greater), we did not consider it for further analysis.

In our analysis, we considered a set of core residues $\mathcal{C}$ and a set of tail residues $\mathcal{T}$. We calculated the average number of tail interactions for a single core residue, $x \in \mathcal{C}$, by counting the number of pairwise interactions, with a lower energy than the threshold, between $x$ and every residue in the $\mathcal{T}$ and dividing by the total number of CTDs:

$$\langle N_x \rangle = \sum_{\text{models}} \sum_{\text{subunits}} \sum_{y \in \mathcal{T}} \delta(x,y) / (N_{\text{subunits}} N_{\text{models}}), \tag{1}$$

where

$$\delta(x,y) = \begin{cases} 1, & \text{if } E(x,y) < -1 \text{ REU} \\ 0, & \text{if } E(x,y) \geq -1 \text{ REU} \end{cases} \text{ and } E(x,y) \text{ is the pairwise energy.}$$

To compute a standard deviation for the average number of interactions with core residue $x$, we used bootstrap resampling as described in *Chaudhury et al. (2011)*. We resampled, with replacement, our set of models for $B = 1,000$ times and re-computed $N'_x$ (same as *Equation 1*, but using the resampled set of models), acquiring a standard deviation: $\sigma_N^2 = \frac{1}{B} \sum_B \left( N'_x - \langle N'_x \rangle \right)^2$.

In addition, we calculated the average energy for each interaction above the threshold (between one core residue, $x \in \mathcal{C}$, and a set of tail residues, $\mathcal{T}$):

$$\langle E_{x:\mathcal{T}} \rangle = \frac{1}{N_{\text{subunits}} N_{\text{models}}} \sum_{\text{models}} \sum_{\text{subunits}} \sum_{y \in \mathcal{T}} E(x,y) \, \delta(x,y), \tag{2}$$

The standard deviation for the interaction energy was computed without bootstrap resampling; the energy has a distribution within a set of models, whereas the presence of an interaction is binary and only varies when the models are resampled. We compute the standard deviation as:

$$\sigma_E^2 = \frac{1}{N_{\text{subunits}} N_{\text{models}}} \sum_{\text{models}} \sum_{\text{subunits}} \left( \sum_{y \in \mathcal{T}} E(x,y)\, \delta(x,y) - \langle E_{x:\mathcal{T}} \rangle \right)^2.$$

Finally, EEC was computed over a set of basic core residues, by multiplying the average tail–core interaction energy by the average number of interactions per model and summing:

$$\text{EEC} = \sum_{\mathcal{B}} \langle N_x \rangle \langle E_{x:\mathcal{T}} \rangle. \tag{3}$$

Standard deviation for EEC was computed by assuming that the standard deviations of the above values are independent: $\sigma_{EEC}^2 = \sigma_E^2 \sigma_N^2 + \sigma_E^2 \langle N_x \rangle^2 + \sigma_N^2 \langle E_{x:\mathcal{T}} \rangle^2$.

## Tail/Core Selections (*E. coli* numbering)

| Species | Core ($\mathcal{C}$) | Basic core ($\mathcal{B}$, for EEC) | Acidic tail ($\mathcal{T}$) |
|---|---|---|---|
| *E. coli* | 1–65 | 3, 16, 17, 19, 47 | 97, 99, 100, 102 |
| *P. aeruginosa* | 1–65 | 3, 5, 16, 17, 19, 47 | 94, 97 |
| *L. monocytogenes* | 1–65 | 2, 16, 17, 19, 35 | 100, 102 |
| *B. subtilis* | 1–65 | 2, 16, 17, 37 | 100, 102 |
| *S. aureus* | 1–65 | 10, 16, 41 | 65, 67, 99, 101, 102 |

## Local concentration of the acidic CTD tip in *E. coli* Hfq

The intrinsically disordered CTD linker was represented by a worm-like chain model (**Kratky and Porod, 1949**) with a statistical chain segment of 35 Å or 10 residues, which is twice the persistence length of 15–20 Å for a random coil polypeptide chain (**Krigbaum and Hsu, 1975**; **Damaschun et al., 1991**; **Damaschun et al., 1993**; **Kellermayer et al., 1997**). The disordered linker region (**Schumacher et al., 2002**; **Beich-Frandsen et al., 2011a**; **Dimastrogiovanni et al., 2014**), was assumed to begin at residue 71 because the first few residues of the CTD tend to pack along the core (**Arluison et al., 2004**). The last five residues of the CTD constitute the acidic tip. The local concentration of the acidic tip on a single CTD was calculated for full-length Hfq (Hfq102) and for Hfq-sCTD:

$$C = \frac{\left( \frac{1}{V_{tail} - V_{core}} \right)}{N_A}. \tag{4}$$

In which $C$ is the concentration of the acidic tip, $V_{\text{tail}}$ is the total volume the acidic tip can access around the center of mass of a single Hfq hexamer, $V_{\text{core}}$ is the inaccessible volume of the Hfq core, and $N_A$ is Avogadro's number. $V_{\text{tail}}$ is estimated as a sphere with radius of 105 Å for Hfq102, and a radius of 70 Å for Hfq-sCTD. $V_{\text{core}}$ is estimated as a cylinder with radius 31.5 Å and height 25 Å (**Sauter et al., 2003**).

## Hfq purification

Untagged *E. coli* Hfq102, Hfq-sCTD, Hfq65, Hfq65-Q35A, Hfq65-K47A, Hfq65-R19D and Hfq65-R16A were over-expressed in *E. coli* BL21(DE3)Δ*hfq::cat-sacB* cells grown in 1 L LB-Miller media (10 g/L Tryptone, 10 g/L NaCl, 5 g/L yeast extract) supplemented with 100 µg/mL ampicillin. Plasmids for over-expression of mutant Hfq proteins were created by site-directed mutagenesis of pET21b-Hfq (**Zhang et al., 2002**). The purification method has been previously described (**Santiago-Frangos et al., 2016**). In brief, resuspended cell lysates of Hfq102 and Hfq-sCTD variants were clarified by heat denaturation and untagged Hfq was purified via Ni²⁺-affinity. Lysates of Hfq65 variants were further clarified by ammonium sulfate precipitation after heat treatment, and

the protein purified by hydrophobic interaction chromatography. Finally, all Hfq variants were purified by cation-exchange chromatography to remove nucleic acids (*Figure 2—figure supplement 1*).

## Nucleic acid preparation

The sequences of RNA and DNA substrates are listed in *Table 2*. Synthetic Target RNAs, molecular beacon (*Panja and Woodson, 2012a*), A18, D16-FAM and R16 have been previously described (*Hopkins et al., 2009*). minRCRB RNA (IDT) was reduced with TCEP (tris(2-carboxyethyl)phosphine) and purified by denaturing PAGE before labeling with Cy3-maleimide (GE Healthcare), as previously described (*Santiago-Frangos et al., 2016*). The extent of labeling was estimated from the absorbance at 260 and 552 nm. The sRNAs ChiX, RprA, DsrA and RyhB and mRNA *chiP* were transcribed *in vitro* as previously described (*Lease and Woodson, 2004*). pUC19 plasmid (NEB) for Hfq binding assays was isolated from transformed DH5α cells (NEB) using Plasmid Maxi kit (QIAGEN) and digested with *EcoR*I (NEB) and *Hind*III (NEB) and purified by phenol-chloroform extraction followed by ethanol precipitation.

## CTD binding and displacement

To measure binding of CTD-FITC, CTDpos-FITC, or BsCTD-FITC peptides (*Table 2*) to Hfq65 or Hfq65 mutants, the fluorescence polarization of FITC-labeled peptide was measured 3 min after the addition of 0–33.3 μM Hfq65. Anisotropy measurements were normalized to the average anisotropy in the absence of Hfq. Samples were prepared in a 100 μL cuvette containing 100 μL 50 mM Tris·HCl pH 7.5, 45 nM CTD-FITC or CTDpos-FITC, at 30°C. Fluorescence polarization with grating correction factor was measured using a Horiba Fluorolog-3 (L-format) with single excitation and emission monochromators at 495 nm and 515 nm respectively (5 nm slit widths). Titrations were performed in duplicate and the curves were fit to a single-site binding isotherm:

$$y = \frac{K_a * x}{1 + (K_a * x)},$$

(5)

in which $K_a$ is the association constant.

Although residue K3 in the NTD was observed to bind the acidic CTD *in silico*, the contribution of K3 to *in vitro* binding could not be determined because neither Hfq65-K3S nor Hfq65-K3Q formed stable proteins.

To measure the displacement of CTD-FITC from Hfq65·CTD-FITC complexes by nucleic acids, samples were prepared in a 100 μL cuvette containing 1 μM CTD-FITC and 1.67 μM Hfq65 hexamer so that roughly 50% of CTD-FITC peptides were bound at the start of the experiment. The polarization of CTD-FITC was measured 3 min after the addition of increasing amounts of RNA or DNA as above. Competition curves were fit to:

$$y = \frac{minY + (maxY - minY)}{1 + \left(\frac{x}{IC_{50}}\right)^n},$$

(6)

where *minY* and *maxY* are the minimum and maximum anisotropy values measured, and $IC_{50}$ is the concentration of nucleic acid which displaced 50% of the bound CTD-FITC from Hfq65.

## RNA binding and annealing

Binding constants for D16-FAM or minRCRB-Cy3 (5 nM) were measured in TNK buffer (10 mM Tris·HCl, pH 7.5, 50 mM NaCl, 50 mM KCl) at 30°C by FAM fluorescence anisotropy as described before (*Hopkins et al., 2009*). To measure the affinity of Hfq for D16-FAM·R16 dsRNA complex, 50 nM of both RNAs were mixed and allowed to equilibrate at 30°C for 10 min before titration with Hfq. Annealing kinetics of molecular beacon (50 nM) to either Target or Target-A18 RNA (100 nM) by 0–200 nM Hfq hexamer, in 1X TNK (10 mM Tris·HCl pH 7.5, 50 mM NaCl, 50 mM KCl) buffer at 30°C was measured by stopped-flow fluorescence spectroscopy as described previously (*Soper et al., 2011*; *Panja and Woodson, 2012b*). Annealing progress curves were fit to single or double-exponential rate equations.

## Anisotropy time-course

To measure RNA binding and release from unlabeled Hfq102, Hfq-sCTD and Hfq65 by anisotropy, the polarization of D16-FAM was recorded every 20 s for $\geq$ 3 min after each addition, as previously described (Santiago-Frangos et al., 2016). Samples were prepared in a 500 µL cuvette containing 50 nM D16-FAM in TNK buffer at 30°C, with additions of 50 nM Hfq102, 50 nM Hfq-sCTD, or 50 nM Hfq65 (binding phase), followed by 50 nM R16 RNA (annealing and release phase), and finally 400 nM of the ssDNA competitor, DNA2 (stimulated release phase). RNA binding and release experiments were done in triplicate for each Hfq variant. The molar fractions of released dsRNA product D16-FAM·R16 ($\chi_{dr}$), remaining ternary complex D16-FAM·Hfq·R16 ($\chi_{hdr}$) and binary complex Hfq·D16-FAM ($\chi_{hd}$) at the end of the 'annealing and release phase' of the experiment were calculated from

$$r_{AP} = (\chi_{dr} * r_{Mdr}) + (\chi_{hdr} * r_{Mhdr}) + (\chi_{hd} * r_{Mhd}), \tag{7}$$

where $r_{AP}$ is the anisotropy measured at the end of the 'annealing and release' phase in the

---

**Table 2.** Sequences of oligomers and sRNAs.

| RNA or DNA oligomers | Sequences (5' to 3') |
| --- | --- |
| Target | GUGGUCAGUCGAGUGG |
| Target-U6 | GUGGUCAGUCGAGUGGUUUUUU |
| Target-A18 | GUGGUCAGUCGAGUGGAAAAAAAAAAAAAAAAAA |
| A18 | AAAAAAAAAAAAAAAAAA |
| R16 | GCACUUAAAAAAUUCG |
| Molecular beacon | FAM-GGUCCCCCACUCGACUCACCACCGGACC-DABCYL |
| D16-FAM | FAM-CGAAUUUUUUAAGUGC |
| minRCRB | Thiol-C6-CUUCCGUCCAUUUCGGACG |
| DNA1 | TATCCGTATGACGTTCCGGACTATGCGGCTAAGGGGCAATCTTTAC |
| DNA2 | TTTTTCAAACTGCGGATGAGACCACATATGTATATCTCCTTCTTAAAGTTAAAC |
| DNA2c | CAAATTGAAATTCTTCCTCTATATGTATACACCAGAGTAGGCGTCAAACTTTTT |
| **Transcribed RNAs** | |
| RprA | gggACGGUUAUAAAUCAACAUAUUGAUUUAUAAGCAUGGAAAUC<br>CCCUGAGUGAAACAACGAAUUGCUGUGUGUAGUCUUUGCCCAU<br>CUCCCACGAUGGGCUUUUUUUU |
| DsrA | gggAACACAUCAGAUUUCCUGGUGUAACGAAUUUU<br>UUAAGUGCUUCUUGCUUAAGCAAGUUUCAUCCCGA<br>CCCCCUCAGGGUCGGGAUUUUUUU |
| RyhB | ggGCGAUCAGGAAGACCCUCGAGGAGAACCUGAAAGCA<br>CGACAUUGCUCACAUUGCUUCCAGUAUUACUU<br>AGCCAGCCGGGUGCUGGCUUUUU |
| ChiX | gggACACCGUCGCUUAAAGUGACGGCAUAAUA<br>AUAAAAAAAUGAAAUUCCUCUUUGACGGGC<br>CAAUAGCGAUAUUGGCCAUUUUUUUU |
| *chiP* | GUAGUCAGCGAGACUUUUCUCAACGCUACUU<br>UUUUAAUUUUUAUUUUUUCGCUGUUCACCUUUG<br>GUGCAGCAAUUUAUACGUCAAAGAGG<br>AUUAACCCAUGCGUACGUUUAGUGGC |
| **Peptides** | |
| CTD-FITC | FITC-Ahx-NNAGGGTSSNYHHGSSAQNTSAQQDSEETE-COOH |
| CTDpos-FITC | FITC-Ahx-NNAGGGTSSNYHHGSSAQNTSAQQRSNKTN-COOH |
| BsCTD-FITC | FITC-Ahx-QLELE-COOH |

DOI: https://doi.org/10.7554/eLife.27049.022

above annealing experiments, $r_{Mdr}$ is the average anisotropy of D16-FAM·R16 complex during the 'stimulated release' phase that is indistinguishable from its anisotropy without Hfq, $r_{Mhdr}$ is the maximum anisotropy of Hfq·D16-FAM·R16 complex from equilibrium binding experiments, and $r_{Mhd}$ is the maximum anisotropy of Hfq·D16-FAM complex from equilibrium binding experiments. Using the conservation of mass, $\chi_{dr} + \chi_{hdr} + \chi_{hd} = 1$, and the relative $K_d$ values for Hfq binding to D16Fl and D16Fl·R16, $\chi_{hd} = K_{rel} * \chi_{hdr}$, yields an expression for the molar fraction of ternary complex:

$$\chi_{hdr} = \frac{(r_{AP} - r_{Mdr})}{(r_{Mdr} * (1 + K_{d\,rel})) + r_{Mhdr} + (r_{hd} * K_{d\,rel})} \tag{8}$$

## Hfq–plasmid DNA binding assays

Samples (10 µL) containing linear pUC19 (0.145 nmol bp), 0–3.333 µM Hfq hexamer, in 40 mM Tris-HCl pH 7.5, 0.14 mM EDTA, 35 mM NH$_4$Cl, 3.7% (v/v) glycerol, 0.05% (w/v) bromophenol blue were incubated at 25°C for 30 min. 2 µL of each reaction was loaded into a sample well of a 15 × 8 cm agarose gel (1.5% (w/v) Seakem LE agarose (Lonza) in 1X TAE (40 mM Tris, 20 mM acetate, 1 mM EDTA, pH 8.0). Electrophoresis was carried out in the cold room (4°C) at 4 V/cm for 6.5 hr. Hfq was dissociated from the bound complexes by soaking agarose gels in 150 mL TBE (89 mM Tris, 89 mM borate, 2 mM EDTA, pH 8.3) and 1 M NaCl, for 30 min at 25°C, at 85 rpm. The gels were washed twice with 150 mL TBE for 10 min, stained with 1X SYBR Gold (Invitrogen) in 150 mL TBE for 45 min, and washed twice with 150 mL TBE for 10 min before imaging on a Typhoon 9410 (GE Healthcare) via excitation at 488 nm and using a 555 nm bandpass 30 emission filter. The fluorescence intensity was measured on a line from the bottom edge of the well through the middle of the lane to visualize the migration of pUC19. For RNA competition experiments, 10 µL samples were prepared as above with 0–2 µM RNA or DNA competitor and 0 or 0.5 µM Hfq102 hexamer.

**Table 3.** Parameters used to calculate molar fractions of D16-FAM complexes.
Values are the mean of at least two independent experiments determined by fluorescence anisotropy. A further description of the parameters and their usage is provided in the Methods.

| Parameter | Anisotropy | Complex/complexes |
|---|---|---|
| $r_{Mdr}$ | 0.0400 | [D16-FAM·R16] |
| Hfq$_{102}$ $r_{Mhdr}$ | 0.2227 | [Hfq102·D16-FAM·R16] |
| Hfq-sCTD $r_{Mhdr}$ | 0.2203 | [HfqsCTD·D16-FAM·R16] |
| Hfq$_{65}$ $r_{Mhdr}$ | 0.2161 | [Hfq65·D16-FAM·R16] |
| Hfq$_{102}$ $r_{AP}$ | 0.0732 | [Hfq102·D16-FAM·R16] + [D16-FAM·R16] + [Hfq102·D16-FAM] |
| Hfq-sCTD $r_{AP}$ | 0.0663 | [Hfq-sCTD·D16-FAM·R16] + [D16-FAM·R16] + [Hfq-sCTD·D16-FAM] |
| Hfq$_{65}$ $r_{AP}$ | 0.1099 | [Hfq65·D16-FAM·R16] + [D16-FAM·R16] + [Hfq65·D16-FAM] |
| Hfq$_{102}$ $r_{Mhd}$ | 0.2195 | [Hfq102·D16-FAM] |
| Hfq-sCTD $r_{Mhd}$ | 0.1985 | [Hfq-sCTD·D16-FAM] |
| Hfq$_{65}$ $r_{Mhd}$ | 0.1935 | [Hfq65·D16-FAM] |
| Hfq$_{102}$ $r_{hd}$ | 0.1332 | [Hfq102·D16-FAM] |
| Hfq-sCTD $r_{hd}$ | 0.1353 | [Hfq-sCTD·D16-FAM] |
| Hfq$_{65}$ $r_{hd}$ | 0.0969 | [Hfq65·D16-FAM] |

DOI: https://doi.org/10.7554/eLife.27049.023

## Acknowledgements

The authors thank Steven Lewis (Cyrus Bio), Susan Gottesman (NCI) and Agata Groszewska (Adam Mickiewicz University, Poznan) for helpful discussion, and Nadim Majdalani (NCI) for the gift of *E. coli* strain NM694 *hfq⁻*.

## Additional information

### Funding

| Funder | Grant reference number | Author |
|---|---|---|
| National Institute of General Medical Sciences | R01 GM120425-01 | Sarah A Woodson |
| National Institute of General Medical Sciences | R01 GM078221 | Jeffrey J Gray |
| National Institute of General Medical Sciences | T32 GM008403-25 | Jeliazko R Jeliazkov |
| National Institute of General Medical Sciences | T32 GM007231-40 | Andrew Santiago-Frangos |

The funders had no role in study design, data collection and interpretation, or the decision to submit the work for publication.

### Author contributions

Andrew Santiago-Frangos, Conceptualization, Validation, Investigation, Visualization, Writing—original draft, Writing—review and editing; Jeliazko R Jeliazkov, Conceptualization, Software, Validation, Investigation, Visualization, Methodology, Writing—review and editing; Jeffrey J Gray, Conceptualization, Software, Supervision, Funding acquisition, Methodology, Writing—review and editing; Sarah A Woodson, Conceptualization, Supervision, Funding acquisition, Visualization, Project administration, Writing—review and editing

### Author ORCIDs

Andrew Santiago-Frangos (iD) http://orcid.org/0000-0001-9615-065X
Jeliazko R Jeliazkov (iD) http://orcid.org/0000-0003-4249-1955
Jeffrey J Gray (iD) http://orcid.org/0000-0001-6380-2324
Sarah A Woodson (iD) http://orcid.org/0000-0003-0170-1987

### Decision letter and Author response

Decision letter https://doi.org/10.7554/eLife.27049.026
Author response https://doi.org/10.7554/eLife.27049.027

## Additional files

### Supplementary files

• Supplementary file 1. Rosetta FloppyTail Protocol. Step-by-step description of FloppyTail simulations, with a fully worked example on *E. coli* Hfq.
DOI: https://doi.org/10.7554/eLife.27049.024
• Transparent reporting form
DOI: https://doi.org/10.7554/eLife.27049.025

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
