## [Decision Letter]

Thank you for submitting your work entitled "Acidic C-terminal domains autoregulate the RNA chaperone Hfq" for consideration by *eLife*. Your article has been favorably evaluated by a Senior Editor and four reviewers, one of whom is a member of our Board of Reviewing Editors. The reviewers have opted to remain anonymous.

Our decision has been reached after consultation between the reviewers. Based on these discussions and the individual reviews below, we regret to inform you that your work will not be considered further for publication in *eLife*.

All of the reviewers are in agreement the results are interesting and provide mechanistic insights into how the natively unstructured region of the Hfq chaperone could act in an auto-inhibitory manner to drive RNA dissociation once duplexes had formed. There is, however, a consensus that the level of conceptual advance in the manuscript is limited, in light of the excellent earlier publication from your laboratory last year on this system. Moreover, the reviewers raise numerous points concerning the data and its interpretation, requiring additional experimental work. We hope that the detailed reviews, appended below, will be of help to strengthen the experimental results so that the manuscript will become suitable for submission to another journal.

Reviewer #1:

In this manuscript, Santiago-Frangos and coworkers study the function of the acidic amino acid-rich CTD tail of *E. coli* Hfq using a combination of modeling, phylogenetic analysis, and biochemical approaches. The main conclusion of this manuscript is that it clarifies the mechanism of Hfq with respect to two competing hypotheses. It has been shown that the CTD can displace RNA from the rim and proximal face of Hfq. This could occur either by a "polymer brush" in which the CTD sweeps the RNA from the surface or by the CTD functioning as a "nucleic acid mimic" to bind the positively charged Hfq core. The authors provide strong evidence for the latter mechanism. The work is clearly described and expertly carried out.

Importantly, the authors propose that the consequence of this mechanism is that it may promote specificity by allowing the CTD to compete with non-specific RNA or DNA binding. In addition, the authors propose that the Hfq CTD is co-evolving with the importance of Hfq functioning as a RNA chaperone in diverse species.

While I think that the specifics of Hfq mechanism would be of interest to a small audience, the implication of these findings for understanding the evolution and use of sRNA regulation in bacteria is great. It is likely that many in the *eLife* readership would be interested in that aspect of the work and it is well-suited for the broad readership of *eLife*.

My major concern is over the role of electrostatics in the CTD interaction. This interaction appears to be governed primarily by negatively charged amino acids (or the phosphate backbone of RNA) interacting with positively charged amino acids. In part my concern stems from the low salt concentrations used in the CTD binding assays (just 50 mM Tris), and I don't believe salt is considered in the FloppyTail predictions. RNA binding and annealing was carried out at higher (but still lower than physiological levels) salt at 10 mM Tris, 50 mM NaCl, 50 mM KCl. Plasmid binding was then carried out at yet a third salt condition.

I'd be curious if CTD binding and displacement still occurs at the 50 mM NaCl/50 mM KCl condition. At minimum, a short discussion of the role of electrostatics in these types of protein/nucleic acid and protein/protein interactions and influence of solution ions would help clarify this issue.

Reviewer #2:

Woodson and coworkers have studied the role of acidic regions in the bacterial RNA chaperone Hfq. By Rosetta modeling and binding studies the authors propose that the C-terminal acidic tail of the Hfq Sm fold interacts with the Sm core domain and thereby affects the RNA chaperone activity, where the Sm core domain normally helps to stabilize sRNA-mRNA duplex formation. It has been previously shown that the Hfq enhances chaperone activity, however, the molecular mechanisms are unclear, and "polymer brush" vs. "nucleic acid mimicking" models have been proposed. The authors propose that the CTD can directly bind to positively charged residues at the rim of Hfq hexamers, in support of the latter model.

The manuscript is clear and well written, the combination of computational modelling and experimental probing is interesting, but has limitations as molecular details are inferred in a more descriptive fashion while a clear experimental picture of the structural details is not provided.

• The authors have used competition binding experiment with FITC-labeled peptides. These are proxies but do not resemble the reality with respect to the fact that in Hfq the CTD will form an intramolecular interaction, which involves much less entropy loss upon binding compared to the peptide titrated in trans. Thus, the apparent affinities and the "quantitative" correlation between experimental and computational K_d_’s has an offset, which should be discussed. Also can the authors exclude any artifacts from the presence of the hydrophobic/aromatic FITC label?

• Preferably, the authors should study the intermolecular interaction with an experimental method. NMR comes to mind as this can reveal the binding site and provide semi-quantitative affinities, for example by comparing Hfq hexamers with and w/o the CTD present and also with peptide titrations in trans (with the caveat mentioned above). These experiments are technically feasible and it is not clear why this has not been considered. The shortened CTD linker constructs could also be well studied in this context and provide an experimental estimation of the local concentration effect by monitoring NMR chemical shifts.

• The model suggests a rather non-specific charged interaction for the binding and biochemical and functional consequences. In the absence of a more precise structural picture for the interactions as suggested above, further support for the non-specific charge-charge interaction and the role of the length in modulating local concentrations should be provided. Artificially designed CTDs could be tested, where the number of negatively charged residues can be controlled and the role of the preceding linker "flexibility" can be explored by replacing the native "linker" with Gly-Ser stretches of varying lengths. These experimental data should be useful to correlate experimental and computational findings and exclude that additional regions in the CTD could play an important role.

Reviewer #3:

In this work, Woodson and colleagues explore the mechanism of the C-terminal domain regulation of Hfq RNA binding. They propose two distinct models and design and carry out experiments that distinguish between them, allowing them to conclude that the CTD displaces RNA from the core by the "nucleic acid mimic" model rather than the "polymer brush" model. The function of the CTD has been a longstanding mystery and source of controversy in the Hfq literature (Updegrove et al., Curr Opin Microbiol 2016; Wagner, RNA Biol 2013). However, the central question that this paper answers is one of mechanism, rather than function. The regulatory function of the CTD had already been established by a previous study published by the Woodson lab (Santiago-Frangos et al., PNAS 2016). This paper expands on those results by providing mechanistic details, but does not offer significantly novel findings regarding the CTD function. Therefore, while this paper contributes to our detailed understanding of Hfq, it is unlikely that it would be of substantial interest to the broad audience of *eLife*.

In addition to this assessment of significance, I have concerns regarding the analysis of the data, which may help the authors if they resubmit to a different journal. Most of the figures, while including a cartoon of a kinetic scheme, do then show data fits that do not accord with the scheme. For example, Figure 2 data are fit to Hill curves, but the described scheme involves binding of a single Hfq variant to a peptide and should therefore use a standard isotherm (Hill coefficient constrained to 1). There are similar ad hoc fits in Figure 3, Figure 4, and 5. The lack of an overall consistent kinetic/thermodynamic framework for computing thermodynamic equilibria and kinetic relaxation curves lowers my confidence in the data analysis and conclusions, including comparisons of contact frequencies in Rosetta simulations to empirical fits and key conclusions based on the shortened CTD construct.

Reviewer #4:

Santiago-Frangos and colleagues describe evidence for a previously unknown role of the C-terminal extension of the Hfq RNA chaperone protein in auto-inhibition. Using a combination of computational molecular modelling and in vitro binding assays, the authors investigate how acidic residues at the C-terminal tip of Hfq interact with the positive nucleic acid binding residues within the core of the Hfq protein. This interaction is predicted to have a role in preventing non-specific interactions with nucleic acids and in accelerating release of annealed RNA products. The manuscript is well written and the conclusions are logical and well presented.

---

## [Author Response]

Reviewer #1:In this manuscript, Santiago-Frangos and coworkers study the function of the acidic amino acid-rich CTD tail of E. coli Hfq using a combination of modeling, phylogenetic analysis, and biochemical approaches. The main conclusion of this manuscript is that it clarifies the mechanism of Hfq with respect to two competing hypotheses. It has been shown that the CTD can displace RNA from the rim and proximal face of Hfq. This could occur either by a "polymer brush" in which the CTD sweeps the RNA from the surface or by the CTD functioning as a "nucleic acid mimic" to bind the positively charged Hfq core. The authors provide strong evidence for the latter mechanism. The work is clearly described and expertly carried out.Importantly, the authors propose that the consequence of this mechanism is that it may promote specificity by allowing the CTD to compete with non-specific RNA or DNA binding. In addition, the authors propose that the Hfq CTD is co-evolving with the importance of Hfq functioning as a RNA chaperone in diverse species.While I think that the specifics of Hfq mechanism would be of interest to a small audience, the implication of these findings for understanding the evolution and use of sRNA regulation in bacteria is great. It is likely that many in the eLife readership would be interested in that aspect of the work and it is well-suited for the broad readership of eLife.

Thank you for this excellent point. We agree that this work has important implications for the molecular evolution of Hfq (and other Sm family proteins), and have revised the Introduction and Discussion sections of the manuscript to emphasize this relationship. In fact, a major question in the field has been whether all members of this large protein family have a similar function. The mechanism presented in this manuscript for the first time gives us a model for predicting how Hfq’s RNA binding and chaperone function has evolved across bacterial genera. Without such a mechanism, there was no basis for extending the results on *E. coli* Hfq to other bacterial Hfqs.

My major concern is over the role of electrostatics in the CTD interaction. This interaction appears to be governed primarily by negatively charged amino acids (or the phosphate backbone of RNA) interacting with positively charged amino acids. In part my concern stems from the low salt concentrations used in the CTD binding assays (just 50 mM Tris), and I don't believe salt is considered in the FloppyTail predictions. RNA binding and annealing was carried out at higher (but still lower than physiological levels) salt at 10 mM Tris, 50 mM NaCl, 50 mM KCl. Plasmid binding was then carried out at yet a third salt condition.I'd be curious if CTD binding and displacement still occurs at the 50 mM NaCl/50 mM KCl condition. At minimum, a short discussion of the role of electrostatics in these types of protein/nucleic acid and protein/protein interactions and influence of solution ions would help clarify this issue.

The reviewer is correct that the tail-core interaction has a large electrostatic component and is therefore sensitive to ionic strength. (RNA binding to Hfq is also sensitive to ionic strength.) This is quite typical of nucleic acid binding proteins such as histone remodelers or retroviral nucleocapsid. For example, the binding of the acidic CTD of HTLV-1 nucleocapsid by its N-terminal domain in *trans* is also weakened by increased salt. In that example, the K_d_ increased from 200 nM at 1 mM NaCl to ~50 µM at 100 mM NaCl (Qualley et al., 2010).

We believe the reviewer is concerned whether the tail-core interactions are stable enough to exist in physiological conditions. To address this question, we had originally measured binding of the CTD peptide to the core of Hfq in higher salt buffers. In 10 mM Tris-HCl, 50 mM K-glutamate), in which chloride is replaced with the more physiologically relevant anion glutamate (Leirmo et al., 1987), the K_d_ increases approximately ten-fold to 22 µM. This titration is now included as Figure 2—figure supplement 2. In 1x TNK (10 mM Tris, 50 mM NaCl, 50 mM KCl) or in 100 mM K-glutamate, we estimated the K_d_ to be ≥ 100 µM although these titrations could not be completed. Because of this, we performed most of our CTD binding experiments in a low salt buffer so that we could saturate binding titrations for all of the mutants and compare their affinities to the wild type protein.

Even in higher salt buffers, the K_d_ values are still similar to or smaller than the estimated local concentration of one acidic CTD tip in *cis*. Molecular crowding in the cell should also favor CTD-core interactions, which we have not accounted for here. Overall, the salt-dependence of the Hfq CTD-core interaction is comparable to what has been observed in other nucleic acid binding proteins. We briefly mention the salt dependence of CTD binding in the first paragraph of the subsection “Acidic CTD specifically binds Hfq rim” and in the first paragraph of the Discussion.

Reviewer #2:[…] The manuscript is clear and well written, the combination of computational modelling and experimental probing is interesting, but has limitations as molecular details are inferred in a more descriptive fashion while a clear experimental picture of the structural details is not provided.

The reviewer brings up an important point, which is that traditional structural methods such as X-ray crystallography or NMR (see below) have so far failed to resolve the details of the CTD interactions. This is precisely because the CTDs are largely disordered in most solution conditions. There are now many examples of disordered or transiently folded segments of proteins that perform important functions, and we think that this flexibility is critical for the normal function of Hfq. We do discuss the details of the interactions in the models and that are supported by our experiments. Our results also suggest how to search for conditions or Hfq sequences in which the CTD is more stably folded.

• The authors have used competition binding experiment with FITC-labeled peptides. These are proxies but do not resemble the reality with respect to the fact that in Hfq the CTD will form an intramolecular interaction, which involves much less entropy loss upon binding compared to the peptide titrated in trans. Thus, the apparent affinities and the "quantitative" correlation between experimental and computational K_d_’s has an offset, which should be discussed. Also can the authors exclude any artifacts from the presence of the hydrophobic/aromatic FITC label?

The reviewer is correct that the absolute values of the energies of CTD-core interactions and the experimentally measured K_d_’s cannot be directly compared. Instead, we compared how much certain amino acid side chains on the core contribute toward CTD binding by calculating an Expected Energetic Contribution (EEC) from the models, which is not a K_d_, and then comparing that to the perturbation of each amino acid substitution to the binding free energy (∆∆G). We hope that this is explained more clearly in our revised text in the last paragraph of the subsection “CTD-bound core residues play a role in RNA annealing”. We agree with the reviewer that the intermolecular peptide binding assay does not capture the effect of having the CTD part of the same polypeptide, which will greatly increase interactions between the CTD and the core of Hfq. It also does not account for interactions between CTDs on the same hexamer. This is now briefly noted in the revised manuscript.

Figure 1 show the results for a control peptide (mutation of the C-terminal tip DSEETE to RSNKTN) that still contains the FITC label yet failed to detectably bind the Hfq65 core. Therefore, the dye is not sufficient for peptide binding in our assay.

• Preferably, the authors should study the intermolecular interaction with an experimental method. NMR comes to mind as this can reveal the binding site and provide semi-quantitative affinities, for example by comparing Hfq hexamers with and w/o the CTD present and also with peptide titrations in trans (with the caveat mentioned above). These experiments are technically feasible and it is not clear why this has not been considered. The shortened CTD linker constructs could also be well studied in this context and provide an experimental estimation of the local concentration effect by monitoring NMR chemical shifts.

The comparative NMR study that the reviewer suggests has already been published in an excellent paper by Beich-Frandsen et al., in 2011. The authors were not able to provide “structural details” of the CTD, owing to the conformational disorder of the CTD itself. However, they did report chemical shift perturbations from the CTD that map to residues on the rim of Hfq, although they did not determine which CTD residues were responsible for these perturbations. We have added this information to the third paragraph of the Introduction. The apparent structural disorder in the CTD in NMR experiments agrees with the transient and heterogeneous nature of the CTD-core interactions in our models, which can be seen in our computational modelling gallery in the new supplement to Figure 1.

The reviewer suggests that NMR experiments to determine the structure of the Hfq CTD should be technically feasible. Segmental labeling and NMR experiments designed to detect transient interactions may indeed provide useful information on this system, but such experiments will be technically challenging. In our own pilot NMR experiments on Hfq with RNA ligands, we also observed that the CTD was largely disordered, with little amide proton chemical shift dispersion. Moreover, WT Hfq forms a 130 kDa dodecamer at NMR concentrations, shortening the T2 relaxation times for core residues. Finally, the CTD peptide itself is prone to aggregation (Fortas et al., 2015), so that peptide titrations at the concentrations required for NMR may not be feasible. For these reasons, we pursued the fluorescence assays described in our paper which require small amounts of CTD peptide.

• The model suggests a rather non-specific charged interaction for the binding and biochemical and functional consequences. In the absence of a more precise structural picture for the interactions as suggested above, further support for the non-specific charge-charge interaction and the role of the length in modulating local concentrations should be provided. Artificially designed CTDs could be tested, where the number of negatively charged residues can be controlled and the role of the preceding linker "flexibility" can be explored by replacing the native "linker" with Gly-Ser stretches of varying lengths. These experimental data should be useful to correlate experimental and computational findings and exclude that additional regions in the CTD could play an important role.

As noted in response to reviewer 1, intramolecular interactions dominated by electrostatics are not uncommon (see HTLV1 nucleocapsid or myosin light chain kinase), and are especially typical in nucleic acid binding proteins where they have been shown to play regulatory functions.

The reviewer suggests some excellent further experiments, and testing artificially designed CTDs was not outside the scope of our imagination. We attempted to make nine additional CTD variants, which in our hands either overexpressed poorly or aggregated. The CTD extrudes from the monomer at a position that could allow it to interact with the polar subunit interface, so it is possible that some CTD variants prevent hexamer assembly. In general, we have found that even small changes to the CTD sequence make Hfq more prone to aggregation.

Reviewer #3:In this work, Woodson and colleagues explore the mechanism of the C-terminal domain regulation of Hfq RNA binding. They propose two distinct models and design and carry out experiments that distinguish between them, allowing them to conclude that the CTD displaces RNA from the core by the "nucleic acid mimic" model rather than the "polymer brush" model. The function of the CTD has been a longstanding mystery and source of controversy in the Hfq literature (Updegrove et al., Curr Opin Microbiol 2016; Wagner, RNA Biol 2013). However, the central question that this paper answers is one of mechanism, rather than function. The regulatory function of the CTD had already been established by a previous study published by the Woodson lab (Santiago-Frangos et al., PNAS 2016). This paper expands on those results by providing mechanistic details, but does not offer significantly novel findings regarding the CTD function. Therefore, while this paper contributes to our detailed understanding of Hfq, it is unlikely that it would be of substantial interest to the broad audience of eLife.

We think that the mechanism should interest a broader audience because it provides a basis for predicting the function of the CTD in other bacterial Hfqs or even in Sm/Lsm proteins. Our proposal that the *E. coli* Hfq CTD acts as a nucleic acid mimic allows us to generalize our model to other members of the family, which would not be possible from the results in the 2016 PNAS paper alone. While we predict that *P. aeruginosa* and *L. monocytogenes* Hfq CTDs perform a similar role as *E. coli* Hfq CTD, our results suggest that *B. subtilis* and *S. aureus* CTDs likely evolved other functions, in the context of their respective Hfq cores. We have revised the Introduction and Discussion to state this general importance of the work more clearly.

Additionally, we provide the first evidence that the Hfq CTD limits its binding to DNA, and we also show that sRNAs are potent inhibitors of DNA binding. Both results call into question the hypothesis that Hfq may regulate various processes in vivo directly via DNA binding (reviewed in Sobrero and Valverde 2012 and Cech et al., 2016).

In addition to this assessment of significance, I have concerns regarding the analysis of the data, which may help the authors if they resubmit to a different journal. Most of the figures, while including a cartoon of a kinetic scheme, do then show data fits that do not accord with the scheme. For example, Figure 2 data are fit to Hill curves, but the described scheme involves binding of a single Hfq variant to a peptide and should therefore use a standard isotherm (Hill coefficient constrained to 1). There are similar ad hoc fits in Figure 3, Figure 4, and 5. The lack of an overall consistent kinetic/thermodynamic framework for computing thermodynamic equilibria and kinetic relaxation curves lowers my confidence in the data analysis and conclusions, including comparisons of contact frequencies in Rosetta simulations to empirical fits and key conclusions based on the shortened CTD construct.

The data analysis is based on previously tested thermodynamic and kinetic models for Hfq binding to RNA (e.g., see Lease and Woodson 2004; Soper and Woodson 2008; Hopkins et al. 2011; Panja and Woodson NAR 2012). We regret that these models were not stated more clearly in this manuscript, and have revised the methods where needed to address this point. We often observe some cooperativity of RNA binding to Hfq (*n* = 1.9 to 2.3). However, we do think that the CTD only occupies one monomer site at a time under the conditions of our titrations, and the Hill coefficients for CTD binding are indeed close to 1. Therefore, we refit all of the binding data to a single-site binding isotherm, which has only a small effect on the K_D_ values (see table below). We agree that this simpler binding model for the CTD is an improvement, and the results from the constrained fits are now shown in our manuscript, along with a revised description in the Materials and methods. Our conclusions do not change.

HfqParameterOriginal fitNew FitHfq65Kd2.862.68Hfq65-Q35AKd2.372.28Hfq65-R16AKd10.1113.40Hfq65-K47AKd5.985.13Hfq65-R19DKd6.046.64Hfq65N1.271Hfq65-Q35AN0.961Hfq65-R16AN0.791Hfq65-K47AN0.871Hfq65-R19DN1.031